# Improving Fairness and Mitigating MADness in Generative Models

## Abstract

Generative models unfairly penalize data belonging to minority classes, suffer from model autophagy disorder (MADness), and learn biased estimates of the underlying distribution parameters. Our theoretical and empirical results show that training generative models with intentionally designed hypernetworks leads to models that 1) are more fair when generating datapoints belonging to minority classes 2) are more stable in a self-consumed (i.e., MAD) setting, and 3) learn parameters that are less statistically biased. To further mitigate unfairness, MADness, and bias, we introduce a regularization term that penalizes discrepancies between a generative model's estimated weights when trained on real data versus its own synthetic data. To facilitate training existing deep generative models within our framework, we offer a scalable implementation of hypernetworks that automatically generates a hypernetwork architecture for any given generative model.

## 1 Introduction

Hypernetworks are neural networks that generate the weights of other neural networks (Ha et al., 2017). Recent work has shown hypernetworks are useful for uncertainty quantification (Rusu et al., 2019), few-shot learning (Sendera et al., 2022), continual learning (von Oswald et al., 2022), among other tasks (Chauhan et al., 2023). To our knowledge, however, no work has evaluated whether generative models trained with hypernetworks produce models that are more fair in representing and generating data belonging to minority classes and more robust to MAD collapse when their output is used as training data.

### 1.1 Motivation: Issues with Maximum Likelihood Estimation

The inspiration for applying hypernetworks to improving fairness and mitigating MADness comes from a realization of the sub-optimiality of Maximum Likelihood Estimation (MLE), one of the most popular techniques for parameter estimation (Johnson, 2013). MLE is used to train most generative model architectures, including variational autoencoders (VAEs) (Pu et al., 2016), normalizing flows (NFs) (Rezende and Mohamed, 2015), diffusion models (Ho et al., 2020), and generative adversarial networks (GANs) (Goodfellow et al., 2014)[1]. Despite MLE's ubiquity, it often produces biased estimators of the underlying true parameters. The most famous example was pointed out by Neyman and Scott (1948), showing that MLE can produce inconsistent results when the number of parameters is large relative to the amount of data (DasGupta, 2008a). In the Neyman-Scott problem, there is not enough data relative to the number of parameters to mitigate the bias, leading to what Neyman called "false estimations of the parameters", or statistics where the stochastic limits are unequal to the values of the parameters to be estimated (Stigler, 2007). This overparameterized regime is precisely where most modern deep learning models are trained (Zhang et al., 2017; Belkin et al., 2019), leading to two problems resulting from this bias: unfairness (resulting from MLE overly prioritizing the generation of datapoints belonging to majority classes), and

---

[1]These observations apply to models trained on a lower bound of the likelihood function, such as the popular ELBO (Kingma and Welling, 2022). Any deep learning model that uses the negative log likelihood as a loss function is performing maximum likelihood estimation (Vapnik, 1999; 1991).

MADness (where models trained on their own output generate poor data (Alemohammad et al., 2023). For an illustration of how bias in MLE penalizes minority datapoints, see Section 5.1, and for an illustration of how the bias in MLE significantly causes MADness, see Section 5.3.

We propose an alternative to MLE that ensures the statistics of parameters estimated from generated data match those estimated from observed data (See Equation 4). Our method, called Penalized Autophogy Estimation (PLE), differs from MLE in that it forces the learned parameters to be recursively stable. Theoretical and emperical results show PLE constrains MLE in a way that removes bias, mitigating the above problems and learning estimators that are fairer and less susceptible to MADness. This recursive debiasing is easily translatable to hypernetworks, where a forward pass maps real or synthetic data to the weights of a downstream network. The difference in statistics between the weights estimated from real versus synthetic data is effectively the bias, which can be penalized in an optimization routine such as stochastic gradient descent. For more details on how this is implemented in a deep learning context, see Section 3.2.

## 1.2 Fairness

The term bias in parameter estimation is distinct from its colloquial usage[2], so to avoid ambiguity, we exclusively use the term "bias" to refer to the statistical bias of an estimator: $b(\hat{\theta}) = \mathbb{E}_{\boldsymbol{X}|\theta}[\hat{\theta}] - \theta$. Recent work has shown that generative models carry and often amplify unbalances present in training data (Zhao et al., 2018). When MLE produces biased estimates of the parameters (as it often does), the parameterized distribution becomes even more concentrated around existing high-probability events[3]. Since probability distributions must integrate to 1, increasing the frequency of some events comes at the expense of decreasing the frequency of others. The other events in this case are those that are less frequent, or belong to minority classes. As one can see in the first row in Figure H.3, biased maximum likelihood estimates eventually collapse towards the mode(s) of the data and thus will underrepresent data away from the mode. As a result, biased estimators will learn distributions where majority-class data is overrpresented and minority-class data is underrepresented, while unbiased estimators will learn distributions that more accurately represent the frequency of minority events.

While recent work has looked at improving fairness in generative models, our work differs conceptually in its focus on removing statistical bias. By removing statistical bias, we avoid over-representing data belonging to majority classes *without needing to specify* any protected attributes or classes. Other approaches are either restricted to a single model type or require data labeled with protected attributes. For instance, FairGAN (Xu et al., 2018) proposes a variant of GANs that requires labeled data with protected attributes, and can only be used for training GANs. Choi et al. (2020) proposes a method that uses two datasets in situations when a smaller dataset may better represent the population ratios, but does not address bias in the learning process itself.

The gradient clipping approach suggested by Kenfack et al. (2022) seeks to improve fairness by biasing the dataset towards uniformity. They write that their goal is "to improve the ability of GAN models to uniformly generate samples from different groups, even when these groups are not equally represented in the training data." This differs from our model in that 1) it actively biases the model to favor more uniform generation of points with different classes to achieve fairness, and 2) requires data labeled with protected attributes, which may not be feasible to expect. Finally, Rajabi and Garibay (2022) suggests a method for generating tabular data whose statistics match a reference dataset. This approach also relies on explicit labels of the protected attribute in the training dataset and is restricted to GANs. Our method can be used for any generative model and requires no labels or information about the protected attribute.

---

[2] The word bias may invoke a normative undertone we wish to be agnostic towards. In fact, some argue a more "fair" model is one where bias is purposely introduced to account for unbalanced classes (Tyler, 1996).

[3] Here we are using the term "event" instead of "data" to be consistent with Kolmogorov's axiomatic treatment of probability spaces; see Section B for more details.

To evaluate the fairness of generative models, we look at the quality of generated samples from models trained on unbalanced datasets containing a majority and minority class ($C_{\text{Maj}}$ and $C_{\text{Min}}$, respectively), where $|C_{\text{Maj}}| \gg |C_{\text{Min}}|$. We define the imbalance ratio as the ratio of datapoints belonging to the majority versus the minority class, $R_I = |C_{\text{Maj}}|/|C_{\text{Min}}|$, and is described by He and Garcia (2009) as the between-class imbalance[4].

We compare this imbalance ratio to the ratio of representation quality from data generated from each class[5]. Let $S$ be a score function which evaluates the representation quality of samples generated from a generative model $M$ (in our experiments in Section 5.1, $S$ is the inverse of the Frechet Inception Distance). $S(M)$ denotes the overall representation quality, which can be broken up into two distinct components: $S(M)_{\text{Maj}}$ and $S(M)_{\text{Min}}$, corresponding to the representation quality of samples from the majority class and minority class, respectively. We introduce a quantity called the *fairness ratio* over a metric $S$, which is defined as

$$R_{\text{Fair}} = S(M)_{\text{Maj}}/S(M)_{\text{Min}}. \tag{1}$$

Values of $R_{\text{Fair}}$ closer to 1 correspond to models that do equally well representing majority and minority datapoints, while values much larger than 1 refer to models that have better performance representing datapoints from the majority class than the minority class. Virtually all variants of empirical risk minimization (including MLE) weight each datapoint equally, and we can thus expect that for a linear $S(M)$,

$$S(M) = \frac{|C_{\text{Min}}|}{|C_{\text{Min}}| + |C_{\text{Max}}|} S(M)_{\text{Min}} + \frac{|C_{\text{Max}}|}{|C_{\text{Min}}| + |C_{\text{Max}}|} S(M)_{\text{Max}}. \tag{2}$$

This implies that $M$ will more accurately model the density around the majority class than the minority class, and thus $S(M)_{\text{Max}} > S(M)_{\text{Min}}$. In other words, even if the training dataset accurately represents the population frequencies of each classes, this relative imbalance often harms performance when looking only at data from the minority class. This is empirically observed with facial classifiers on unbalanced data (Buolamwini and Gebru, 2018) and is seen for vanilla generative image models in Table 1.

In principle, weighing each datapoint equally corresponds to a process considered to be procedurally fair (Tyler, 1996), despite the fact that the representation quality for samples in the minority class may be far worse than those in the majority class. As a result, one can argue that $R_{\text{Fair}} = R_I = |C_{\text{Maj}}|/|C_{\text{Min}}|$, represents results from a procedurally fair training process, where each datapoint is treated equally[6]. However $R_{\text{Fair}} \gg R_I$ is clearly unfair, as it penalizes the generation of minority data far more than would be expected by its underrepresentation in the training set. Comparing $R_I$ to $R_{\text{Fair}}$ in practice shows standard generative model training is far worse at representing data from minority classes than $R_I$ would suggest. We observe that removing statistical bias when estimating parameters leads to increased performance representing data from minority classes. As a result, hypernetwork training and its bias-removal properties leads to more fair outcomes, as we show in Section 5.1.

### 1.3   Model Autophagy Disorder (MADness)

Recent work has shown that models trained on their own output, a process called a self-consuming loop, progressively decrease in quality (precision) and diversity (recall) (Alemohammad et al., 2023). Researchers call this phenomenon "going mad" or simply "mad cow," after bovine spongiform encephalopathy (BSE), the medical term for mad cow disease[7]. This phenomenon has become a growing concern for the machine learning community due to the availability and ubiquity of synthetic data (Nikolenko, 2021). It often occurs with large

---

[4]Note that an unbalanced dataset is not necessarily *biased*; the class imbalance in an unbalanced dataset may accurately represent the true ratio of majority to minority datapoints in the population.

[5]This task is not necessarily conditional; none of our experiments use conditional generation. Rather this refers to the generation quality of samples that are *classified* as belonging to either class.

[6]Unconditional generative models have no "knowledge" of protected attributes.

[7]Bovine spongiform encephalopathy (BSE) is a neurological disorder believed to be transmitted by cattle eating the remains of *other* (infected) cattle (Prusiner, 2001).

language models (LLMs) such as ChatGPT: LLMs trained on their own output suffer in diversity, eventually collapsing to a single point (Briesch et al., 2023).

Given the popularity and availability of these models, it is almost inevitable that future LLMs will be trained on a corpus containing at least some (if not much) synthetic data, implying that future versions of ChatGPT and similar LLMs may be subject to diversity collapse. We show that the presence of estimator bias worsens this phenomenon, and we propose a method of removing this bias for deep generative models in Section 4, effectively slowing down this collapse. Biased maximum likelihood estimates of distribution parameters also exhibit MADness, which is shown for several popular distributions in Figure 4.

## 2 BACKGROUND

### 2.1 UNBIASED ESTIMATION

Bias correction literature is relevant to generative model training since MLE often produces biased results (for a brief overview of generative models, see Section C). There are many specific methods for reducing or eliminating bias in parameter estimation problems (Singh and Singh, 1993), and bias correction methods have been proposed for generalized linear models (Cordeiro and McCullagh, 1991), autoregressive-moving average (ARMA) models (Cordeiro and Klein, 1994), convex regularized estimators (Bellec and Zhang, 2021), diffusion processes (Tang and Chen, 2009) and specific distributions of interest (Singh et al., 2015; Cribari-Neto and Vasconcellos, 2002). While bias correction is often done via bootstrapping (Efron, 1979; Jiao and Han, 2020) or jackknifing (Quenouille, 1956; 1949), our work is most closely related to using parametric bootstrapping for bias correction (Kosmidis, 2014). Parametric bootstrapping uses synthetic or generated samples to estimate and remove empirical bias (Hall, 1992), and has been described by Efron (2012) as linking Bayesian and frequentist perspectives. We explain how our estimation procedure also links Bayesian and frequentist perspectives in Section A.

Unlike much existing work, our method described in Equation 4 does not require assuming the bias is additive or multiplicative (Ferrari and Cribari-Neto, 1998). Furthermore, recent work has shown the penalizing the square of estimated bias produces asymptotic minimum-variance unbiased estimators (MVUEs) and asymptotically hits the Cramer-Rao bound (Diskin et al., 2023). This is related to our relaxation of Equation 4 in Section 4.

### 2.2 PARAMETER ESTIMATION AND MODEL AUTOPHAGY

Given data $\mathbf{X}$ drawn from a parameterized distribution $P(\mathbf{X}; \theta)$, we form an estimator $\hat{\theta}$ of the generative model's parameters $\theta$. As shown in Figure 1, the model's parameters $\theta$ are estimated by some function of the observed data $\hat{\theta} = H(\mathbf{X})$. MADness (the collapse of generated quality) arises when the estimated parameters are used in the generative model to produce a new dataset $\widehat{\mathbf{X}}$ and this dataset is again used to estimate the model's parameters.

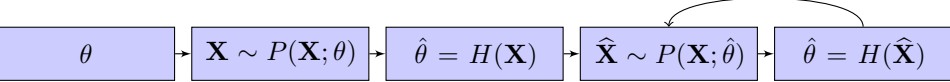

Figure 1: The self-consuming parameter estimation loop.

## 3 AUTOPHAGY PENALIZED LIKELIHOOD ESTIMATION (PLE)

### 3.1 THEORETICAL FORMULATION

PLE involves adding a constraint to the maximum likelihood estimator to force it to take into account other possible models that could have generated the data. To illustrate the process, the conceptual model consists of several steps:

1. Choose a parametrization $P(\mathbf{X}; \theta)$ for our data-generation model where $\mathbf{X} = [\mathbf{x}_1, \ldots, \mathbf{x}_n]$, with each $\mathbf{x}_i \sim P(\mathbf{x}; \theta)$, $i = 1, \ldots, n$ are I.I.D. samples from the generative model parameterized by $\theta$.

2. Choose a function $H(\cdot)$ that deterministically produces an estimate of $\theta$ from $\mathbf{X}$: $\hat{\theta} = H(\mathbf{X})$.

3. Generate the candidate set of estimators $C = \left\{ \hat{\theta} \text{ s.t. } \mathbb{E}_{\mathbf{X}, \mathbf{Y}}[H(\mathbf{Y}) - H(\mathbf{X})] = 0 \right\}$, over all *possible* data $\mathbf{Y}$ generated by $P(\mathbf{Y}; H(\mathbf{X}))$.

4. Choose the estimator from $C$ that maximizes the likelihood function: $\hat{\theta}_{\text{PLE}} = \arg\max_{\hat{\theta} \in C} P(\mathbf{X}; \hat{\theta})$.

This process can be summarized as a constrained maximum likelihood estimation problem:

$$\hat{\theta}_{\text{PLE}} = H^*(\mathbf{X}), H^* = \arg\max_H P(\mathbf{X}; \hat{\theta}) \text{ s.t. } \mathbb{E}_{\mathbf{X}, \mathbf{Y}}[H(\mathbf{Y}) - \hat{\theta}] = 0. \tag{3}$$

Since $\theta = H(\mathbf{X})$, we can also write this constraint fully in terms of $H$:

$$\hat{\theta}_{\text{PLE}} = H^*(\mathbf{X}), H^* = \arg\max_H P(\mathbf{X}; H(\mathbf{X})) \text{ s.t. } \mathbb{E}_{\mathbf{X}, \mathbf{Y}}[H(\mathbf{Y}) - H(\mathbf{X})] = 0. \tag{4}$$

Equation 4 is essentially MLE with an equality constraint enforcing the statistics of synthetic data, $\mathbf{Y}$ to match that of the observed data $\mathbf{X}$. The key here is that the same estimation procedure $H$ (which will eventually be a hypernetwork in our experimental setup) that produces $\theta$ from $\mathbf{X}$ can *also* be used to estimate parameters from synthetic data $\mathbf{Y}$. When this synthetic data is drawn from a distribution parameterized by $\theta$ itself, any change in estimated parameters (in expectation) becomes a proxy for estimator bias.

The difference between Equation 4 and traditional debiasing techniques such as those discussed in Section 2.1 is that PLE is recursive and adapts to the observed data $\mathbf{X}$. As a result, it does not depend on a specific choice of $H$ and can be used in general to debias of estimators when a closed form expression is available (See Sections E.4 and F.3) and, with a few modifications, to debias the training generative models, as discussed in the next several sections.

## 3.2 COMPUTATIONAL IMPLEMENTATION

Evaluating the constraint in Equation 4 is computationally intractable because the expectation requires integrating over all *possible* synthetic data. We make this problem tractable by first turning the constrained optimization problem into an unconstrained one via a Lagrangian relaxation:

$$H^* = \arg\max_H P(\mathbf{X}; H(\mathbf{X})) + \lambda \mathbb{E}_{\mathbf{X}, \mathbf{Y}}[H(\mathbf{Y}) - H(\mathbf{X})].$$

Here, $\lambda$ is a hyperparemeter called the PLE penalty which penalizes differences in the statistics of parameters estimated from training versus synthetic data. For our experiments, we set $\lambda = 0.1$ based on the empirical ablation experiments (See Section J for more details). To make this expression even more tractable, we can estimate this expectation above via a parametric bootstrap with $m$ synthetic samples. Let $\hat{\mathbf{Y}} = [\mathbf{y}_1, \ldots \mathbf{y}_m]$ represent $m$ samples[8] of synthetic data drawn from a distribution parametrized by $H(\mathbf{X})$.

$$H^* = \arg\max_H P(\mathbf{X}; H(\mathbf{X})) + \frac{\lambda}{m} \sum_{i=1}^m \left| H(\hat{\mathbf{Y}}_i) - H(\mathbf{X}) \right| \tag{5}$$

The above equation is how PLE can be used in principle to train generative models, and it differs from traditional, MLE-based training in two ways. The first is that rather than maximizing the likelihood function $P(\mathbf{X}|\theta)$ with respect to the parameters $\theta$, we maximize the likelihood with respect to a hyper learning task $H$. This hyper learning task is designed to generate parameters $\theta$ when given training data $\mathbf{X}$ *or* synthetic data $\mathbf{Y}$. The second is our introduction of the PLE penalty $\frac{\lambda}{m} \sum_{i=1}^m \left| H(\hat{\mathbf{Y}}_i) - H(\mathbf{X}) \right|$, which penalizes hyper-learning mechanisms that recursively differ in estimated parameters.

---

[8]These samples need not be individual points; they can each be $n-$dimensional or share the batch size of $\mathbf{X}$.

## 4 Implementing $H$ with Hypernetworks

Solving the optimization problem in Equation 4 in real-world applications is intractable due to two main computational bottlenecks. First, the operator $H$ that produces the weights of the generative model, $P(\mathbf{X}; \theta)$, from training data, $\mathbf{X}$, is not an explicit operator. Instead, it is usually an optimization routine, (i.e., training of a generative model given data). As a result, evaluating the PLE constraint in Equation 4 involves solving an inner optimization problem that trains a secondary generative model on synthetic data. This is clearly intractable as it requires training a new generative model at *every* iteration. Second, it is unclear how the PLE constraint can be strictly imposed.

To address these challenges, we propose parameterizing $H$ as a hypernetwork (Ha et al., 2017) denoted by $H_\phi$, i.e., a neural network that is trained to predict the weights of another neural network. In our case, we use $H_\phi$ to learn the parameters of $P(\mathbf{X}; \theta)$: $H_\phi$ takes as input training data and predicts the weights of a generative model that approximates the distribution of the training data. This downstream generative model is never explicitly trained via backpropagation; rather its weights are set via $H$. This allows the PLE constraint to be tractably evaluated by a sample of data through $H_\phi$. To address the second challenge, we relax the optimization problem in Equation 4 so the constraint is turned into a penalty term.

Inspired by the form of $H$ obtained analytically for some simple distributions in Appendix E.1, and to impose permutation invariance — with respect to ordering of data points — we propose the following functional form for the hypernetwork $H_\phi$ following Radev et al. (2022):

$$H_\phi(\mathbf{X}) := h_\phi^{(2)} \left( \frac{1}{n} \sum_{i=1}^{n} h_\phi^{(1)}(\mathbf{x}_i) \right), \tag{6}$$

where $h_\phi^{(1)}$ and $h_\phi^{(2)}$ are two different fully-connected neural networks and $\mathbf{X} = [\mathbf{x}_1, \dots, \mathbf{x}_n]$ is a set of data samples. This permutation invariance is similar to the permutation symmetry assumed in U-statistics (DasGupta, 2008b; Hoeffding, 1948). More recommendations for choosing $H$ based on theoretical considerations can be found in Section H.1.

While more expressive architectures can be used, in our experiments we found it sufficient to choose $h_\phi^{(2)}$ to be a set of independent fully-connected layers such that for any layer in the target generative model, we predict its weights via applying an independent fully-connected layer to the intermediate representation $\frac{1}{n} \sum_{i=1}^{n} h_\phi^{(1)}(\mathbf{x}_i)$. This functional form has several advantages: (i) it is permutation invariant, which is required since the weights of the generative model $P(\mathbf{X}; \theta)$ do not depend on the order of data points $\mathbf{X} = [\mathbf{x}_1, \dots, \mathbf{x}_n]$; (ii) it can deal with an arbitrary number of data points, which enables batch training and in turn allows for evaluating $H_\phi$ over a large number of data points that otherwise would not fit into memory; and (iii) the inner sum in the functional form of $H_\phi$ and the proposed architecture for $h_\phi^{(2)}$ (a set of independent linear layers) are highly parallelizable, allowing us to train existing generative models with minimal computational overhead.

## 5 Experiments

For each experiment, we use one or two NVIDIA Titan X GPUs with 12 GB of RAM. The time of execution for each experiment varies from a few minutes to 14 hours per model trained. For the experiments with multiple models trained, since we must train the models sequentially, the time of execution of the whole experiment is just the number of models times the time necessary to train one model.

### 5.1 Fairness Experiments

As discussed in Section 1.2, we evaluate the quality of generated samples on datasets with varying imbalance ratios, $R_I$. As is common in the literature, we use the Frechet Inception

Table 1: FID for Minority and Majority data, trained on MNIST with a Variational Autoencoder (VAE). Note that this task is an unconditional generative task; the generated images are sent through a pretrained classifier to determine the corresponding class. FID is calculated using the weights from ResNET-18 on the MNIST training set. Lower FID is better, as this corresponds to generated images being more similar to the images in the training set; example images can be seen in Section K. The VAE consists of an encoder and decoder, each with 5 layers containing a fully connected layer with batchnorm and leaky Relu. This was trained on a single CPU in several hours. The hypernetwork architecture consists of hidden sizes of 32,64, and 96, which takes several hours to train on a single CPU, taking a few hours longer than the baseline. The majority class was the digit 3 and the minority class was the digit 6 (this choice done randomly, as the goal of this experiment is to show the effect of our method on the minority class, which depends primarily on the frequency of occurrence and not the class itself.), with the ratio of majority to minority datapoints used for training shown in the table.

| Model | Majority Class FID | Minority Class FID | $R_{\text{Fair}}$[10] | Overall FID |
|---|---|---|---|---|
| $R_I = C_{\text{Maj}} : C_{\text{Min}} = 2 : 1$ | | | | |
| VAE | 0.6666 | 1.0544 | **1.5817** | 0.6670 |
| Hyper-VAE (Ours) | **0.4456** | **0.9671** | 2.1703 | **0.5227** |
| $R_I = C_{\text{Maj}} : C_{\text{Min}} = 5 : 1$ | | | | |
| VAE | **0.4147** | 2.6167 | 6.3097 | 0.6313 |
| Hyper-VAE (Ours) | 0.4572 | **2.0532** | **4.4904** | **0.6299** |
| $R_I = C_{\text{Maj}} : C_{\text{Min}} = 10 : 1$ | | | | |
| VAE | **0.3060** | 3.9760 | 12.993 | **0.4803** |
| Hyper-VAE (Ours) | 0.4482 | **2.9806** | **6.650** | 0.5856 |
| $R_I = C_{\text{Maj}} : C_{\text{Min}} = 20 : 1$ | | | | |
| VAE | **0.2324** | 9.9098 | 42.641 | **0.6116** |
| Hyper-VAE (Ours) | 0.4122 | **6.03186** | **14.633** | 0.8603 |

Distance (FID) (Heusel et al., 2017a) to evaluate the quality of generated images. Our score metric $S$ is the inverse of FID since smaller distances correspond to higher quality generated images. This is so it can be used as an appropriate score metric $S$ in the fairness ratio $R_{\text{Fair}}$[9].

While it may not be reasonable to expect $R_{\text{Fair}} = 1$ in cases when $|C_{\text{Maj}}| \gg |C_{\text{Min}}|$, we show that models trained with hypernetworks plus a PLE penalty have values of $R_{\text{Fair}}$ much closer to 1 than those trained with MLE. Furthermore, our experiments suggest models trained with PLE have $R_{\text{Fair}} < |C_{\text{Maj}}|/|C_{\text{Min}}|$, implying that PLE helps the generation of minority data beyond what the class imbalance would predict. The results for models trained with Hypernetworks versus MLE based training are shown in Table 1, showing that hypernetwork training produces results that are more fair. These results become mxore pronounced as the classes become more and more imbalanced (as $R_I = |C_{\text{Maj}}|/|C_{\text{Min}}|$ increases)

---

[9]Recall from Section1.2 that $S$ compares the representation *quality* of samples from the majority to the minority classes.

[10]Closer to 1 is better.

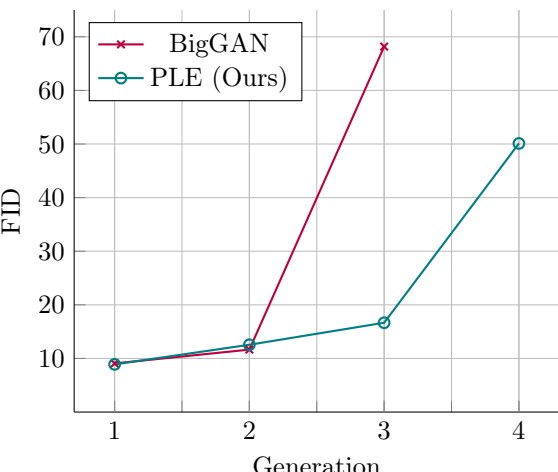

Figure 2: PLE is more stable and outperforms the baseline as we train models on their own outputs (MADness). This plot shows the generation versus FID for BigGAN trained on CIFAR-10. The baseline (labeled BigGAN) uses normal BigGAN training and collapses after only three generations. On the other hand, our method is very stable and only sees a slight increase in FID over the course of the three generations.

## 5.2 Illustrative Example: Unbalanced Gaussian Mixture Model

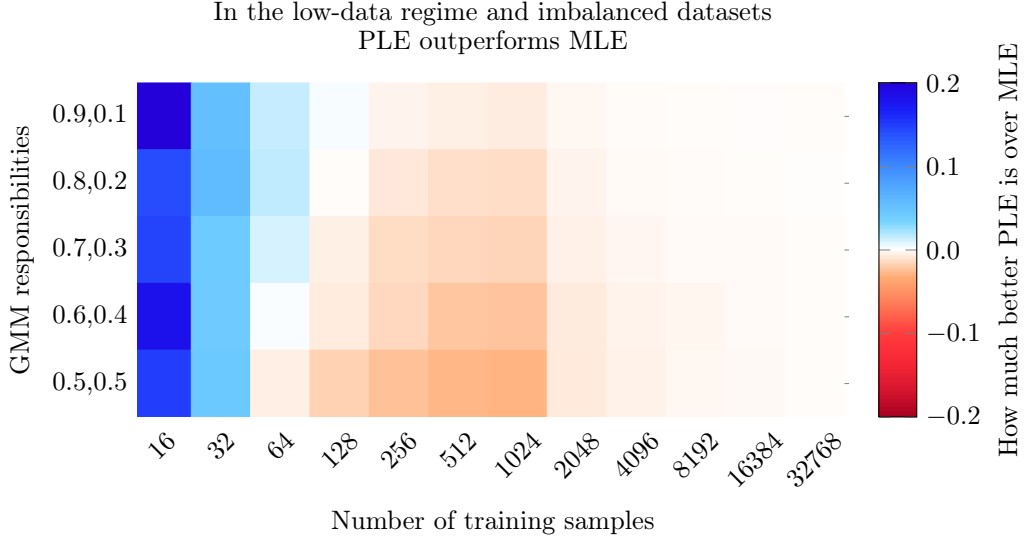

Figure 3: Comparison of KL divergence differences, i.e., $\mathbb{D}_{KL}(q_{\mathrm{MLE}} \| p_{\mathrm{GMM}}) - \mathbb{D}_{KL}(q_{\mathrm{PLE}} \| p_{\mathrm{GMM}})$, for estimating GMM parameters. Positive values indicate scenarios where PLE outperforms MLE, particularly in imbalanced datasets and low-data regimes.

To showcase the benefits of hypernetwork training when dealing with imbalanced datasets, we estimate the weights of a one-dimensional Gaussian mixture model (GMM) with two components that have considerable overlap. The means of the two Gaussian components are 0.0 and 2.0 and the variances are both equal to 1.0. We vary the responsibility vector such that the contribution of one of the two Gaussian components is decreased. This makes

estimating the true GMM parameters challenging, especially in low-data regimes and as the GMM becomes more imbalanced.

We estimate the GMM weights using expectation maximization (EM), the gold standard MLE approach, and compare the results to the estimated GMM via hypernetwork training. We rely on `sklearn` for estimating the GMM via the EM approach, choosing an optimization tolerance smaller than floating point precision, and setting the maximum number of iterations to $10^5$ (we did not observe improvement by increasing this number). For hypernetwork training, we define $H$ to predict the means, variances, and the responsibility vector from the training data in one step. The architecture of $h_\phi^{(1)}$ contains three fully-connected layers with hidden dimensions of 8 and ReLU activation functions. We design $h_\phi^{(2)}$ in a similar manner, with the only difference being the last layer, which contains three branches. Each branch includes a linear layer aimed at predicting the mean, variance, and responsibility vector of the GMM. We ensure the predicted variance is positive by using a softplus activation function and ensure the predicted responsibilities are positive and sum to one by using a softmax function. The objective function in the PLE case is to maximize the likelihood of observing the training data under a GMM model whose weights are predicted via the hypernetwork, while also including the PLE constraint as a penalty term with a weight of 0.1.

To compare the performance between MLE and PLE, after optimization, we calculate the KL divergence between the estimated and the ground truth GMMs using $10^5$ samples. We repeat the GMM estimation for 100 different random seeds and average this quantity. Figure 3 illustrates the difference between the KL divergence between PLE and the true GMM minus the KL divergence between MLE and the true GMM, i.e., $\mathbb{D}_{KL}(q_{\mathrm{MLE}} \| p_{\mathrm{GMM}}) - \mathbb{D}_{KL}(q_{\mathrm{PLE}} \| p_{\mathrm{GMM}})$. The negative values (shown in blue) correspond to settings where PLE outperforms MLE (in terms of KL divergence). We observe that for imbalanced GMMs, and when the training data size is smaller, PLE clearly outperforms MLE. In addition, as expected, as the number of training samples goes to infinity, the performance of PLE and MLE become similar.

## 5.3 MADness Experiments

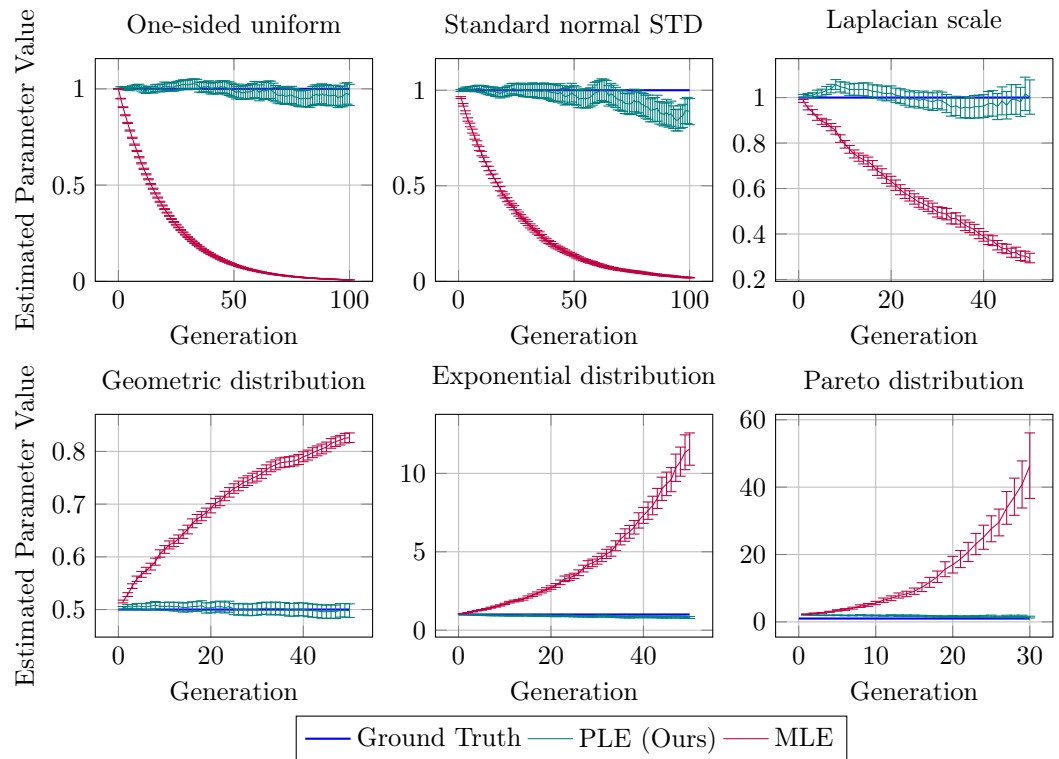

Figure 4: MLE vs PLE Estimates of the parameters of various distributions. Notice how MLE collapses into MADNess much faster than PLE. More details can be found in Section H.3

Our experiments in this section show that models trained with PLE (either analytically or via our hypernetwork approach) are less susceptible to MADness (Model Autophagy Disorder (Alemohammad et al., 2023)) than models trained with MLE. In the following experimental setups, parameters are estimated from fully synthetic data, generated either from a model trained on the ground truth data, or the synthetic data from a previous generation.

An illustrative example involves estimating the parameter $a$ of a one-sided uniform, $U[0, a]$. While the MLE and PLE of $a$ are similar in their closed-form expression (See E.1 for more details), the two quickly diverge and the MLE collapses to 0 as a new estimate is produced from data generated from the previous estimate. MLE's collapse is due to its bias: this bias degrades the estimation quality after each generation, as seen in Figure 4. Section E.3 shows the result of PLE when using a linear function class, while Section E.4 shows the result of PLE when using a linear function of the $n$th order statistic. These two PLE results agree in expectation and are both unbiased. Similar comparisons of MLE vs. PLE on various distributions are shown in Figure 4.

We also trained BigGAN (Brock et al., 2018) on CIFAR-10 (Krizhevsky et al., 2009) and observe a similar result. Specifically, the BigGAN [11] data generation collapses after 3 iterations whereas our PLE method does not. We measure performance here using FID (Heusel et al., 2017b), which does not change much for our method over the course of 3 generations yet completely explodes for the BigGAN baseline (see Figure 2). Additionally, both the baseline and the PLE took about the same time, 14 hours on two GPUs.

---

[11]We use https://github.com/ajbrock/BigGAN-PyTorch

## 6 Conclusion

Hypernetwork training is promising for the training of generative models due to its removal of bias, its improvement of representation fairness, and its mitigation of MADness. Future work can focus on additional applications of hypernetwork training, such as mitigating overfitting due to the ability of hypernetworks to quantify uncertainty. Since hypernetwork training involves sampling the generative model to evaluate the penalty, future work is needed to allow tractable training of diffusion models, which are expensive to sample from[12]. Furthermore, future work can explore guidelines for setting and scheduling the PLE penalty $\lambda$ during training. By combining unbiased statistical estimation methods with deep learning, we believe we can make artificial intelligence more fair and stable.

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

## A   PLE as Bayesian *and* Frequentest Estimation

In the field of statistics, there is a divide between Bayesians and frequentists. The Bayesian approach sees the fixed (and unknown) parameters as random variables (Wakefield, 2013). In this case, the *data* is fixed while the parameters are random (Fornacon-Wood et al., 2022). The benefit of the Bayesian approach is its ability to find optimal estimators: because parameters are mapped to a probability measure, they can be compared and a maximum conditioned on data can be found (when it exists). The drawback of Bayesian estimation is that it requires an accurate prior; if the prior is inaccurate, the estimator may ignore the data and produce a biased result (Dias et al., 2014). Additionally, detractors point out that the choice of prior (and the form of the posterior itself) is often subjective Gelman (2008).

The frequentist approach, on the other hand, evaluates a hypothesis, which corresponds to a specific choice of parameters, by calculating the probability of observed data *under* this hypothesis. In this case, the *hypothesis* is fixed while the data is a random variable. As one statistician writes:

> As the name suggests, the frequentist approach is characterized by a frequency view of probability, and the behavior of inferential procedures is evaluated under hypothetical repeated sampling of the data (Wakefield, 2013).

Since the true parameters in an estimation problem are often assumed to be non-random, the frequentist is correct to point out that uncertainty in estimation is usually *epistemic*, not *ontic*. This observation, as true as it may be, comes at a cost: there becomes no obvious way of *choosing* optimal parameters. For the Baysian, the a posteriori distribution serves as a goodness measure that tells us how well a given hypothesis fits the observed data. The frequentist, on the other hand, has no such measure: since the parameters are fixed, we are unable to make probability statements about the parameters given the observed data (Wagenmakers et al., 2008). This leads to the most fundamental limitation of frequentist inference: it does not condition the observed data (Wagenmakers et al., 2008). Of course, a frequentist can make choices based on the observed data (such as a particular choice of a kernel function in Kernel Density Estimation), however the Bayesian will often point out such assumptions are contrived (Hájek, 2007; Romeijn, 2022).

The promise of a hybrid view comes from an acknowledgement of the benefits and shortcomings of each approach. As Roderick Little suggests, "inferences under a particular model should be Bayesian, but model assessment can and should involve frequentist ideas (Little, 2006)[13]" The Bayesian is correct to point out that we are estimating our parameters from

---

[13]See also (Gelman, 2008; Rubin, 1984).

observed *data*, so there will be uncertainty in the parameters themselves. The frequentist, on the other hand, is correct to point out that the randomness in estimation results from the data itself and not true parameters we wish to estimate. This is what causes the problem in Section E.1, and with bias in MLE more generally: while the estimated parameters are random under the data (which is true), the randomness of the data *under* the ground truth model (which are used to estimate the parameters) is ignored.

PLE bridges this gap: it treats the true parameters as nonrandom, while acknowledging that the estimated parameters are random *because* the data is random. It incorporates the uncertainty of the estimation process from the uncertainty of the data in a given model class. As a result, in virtually all cases, the strength of estimation uncertainty is related to the number of datapoints - for consistent models, as $n \to \infty$, the mutual information between the data and the true parameters increases.

## B  Sampling, Randomness, and Modal Logic

Probability theory is a way of saying how "likely" something is to happen. A probability space is a $3-$element tuple, $S = (\Omega, F, P)$, consisting of

- A **sample space** $\Omega$ which is a (nonempty) set of all possible outcomes $\omega \in \Omega$
- An **event space** $F$, which is a set of events $f$ which themselves are sets of outcomes. More precisely, $F$ is a $\sigma$-algebra over $\Omega$ (Stroock, 2010).
- A **probability function** $P$ which assigns each $f \in F$ to a *probability*, which is a number between 0 and 1 inclusive.

This triple must satisfy a set of probability axioms to be considered a legitimate probability space (Papoulis and Pillai, 2002). An event with probability 1 is said to happen *almost surely* while an event with zero probability is said to happen *almost never*. Note that not all zero probability events are logically impossible (The probability of *any* outcome on a continuous probability distribution is 0, even after one *observes* such an outcome). Impossible events are thus those not contained within $F$; these are assigned probability of zero by definition.

This idea of probability is closely related to the idea possible worlds, other ways the world *could have been*. We use the semantics of modal logic to write the different "modes" of truth, including *necessary* propositions ($\Box x$), *possible* propositions ($\Diamond y$) and *impossible* propositions ($\neg \Diamond z$) (Garson, 2024). Consider a fair dice roll on a six-sided die, where each outcome is the corresponding number on top of the die (from 1-6). Let $f_{n<10}$ be the event that one rolls a number less than 10. Since all possible outcomes have a value less than 10, we say it is *necessary* that one rolls a number less than 10, or $\Box f_{n<10}$. Let the $f_4$ be the event that one rolls a 4. It is *possible* that one rolls a 4, so we say $\Diamond f_4$, however it is not necessary, because one could have rolled a 5 instead. It is *impossible* to roll a 7, so we say $\neg \Diamond$roll a seven. Rolling a seven is not an event because it does not exist in $\Omega$; 7 is not a legitimate outcome as constructed[14].

An outcome $x \sim P(\theta)$ from a probability distribution $P$ cannot feature all possibilities from the distribution except in the case of a trivial distribution (which has no randomness). For nontrivial distributions, or ones with infinite support, there will always be at least some other outcome with nonzero probability $\tilde{x}$ that *could have happened* if our observed outcome was different. Semantically, we say there is a *possible world* in which $\tilde{x}$ happens as long as $\tilde{x} \in F$ (Menzel, 2023).

## C  Background on Generative Models

Generative models use data to estimate an unknown probability distribution, generating "new" data by sampling from the estimated distribution. A good generative model generates data whose statistics match that of the observed data used to train the model (Schwarz et al., 2021). From a Bayesian perspective, training a generative model amounts to using observed

---

[14]It is common to notate such events using the empty set.

data to update the estimated parameters that are assumed to give rise to the data. The architecture of a generative model can be viewed as a prior on the estimated distribution, constraining the overall shape of the distribution itself (Ulyanov et al., 2018).

Of course, the data are merely samples of (what is usually assumed to be) an underlying stationary stochastic process with fixed parameters. Generative models seek to find these fixed parameters so as to determine the shape and location of the distribution from which our data are assumed to have been drawn. Randomness is introduced in the estimation process in two ways: 1) Each datapoint is random and 2) the sampling process itself contains randomness based on the relationship between the given datapoints. More data implies a given model can better discriminate between competing parameter choices, decreasing estimation uncertainty and thus estimation randomness. In many cases, an infinite amount of data would lead to an estimator that is completely deterministic, as there would be enough data to remove any estimation uncertainty.

## D    Maximum Likelihood Estimation

### D.1    Model Estimation

Model estimation involves estimating an unknown probability function $P$ (one of the elements of a probability triple $(\Omega, F, P)$ (Kolmogorov and Bharucha-Reid, 2018)) from samples, each sample an outcome from its sample space (see B for more details). We notate this as $\mathbf{X} = [\mathbf{x}_1, \ldots \mathbf{x}_n]$, $\mathbf{x}_i \in \Omega$, $i = 1, \ldots, n$. In most cases (and in all practical cases), the number of samples we observe is finite ($|X| \in \mathbb{Z}^+$). Estimating $P$ from a finite number of samples is made difficult by the fact that we not only want to estimate the probability of the observed events (corresponding to the samples), but also estimate the probability of unobserved events that *could have happened*. Since we do not directly observe all events or the probability function itself, we must estimate the elements of a probability triple ($\Omega$, $F$ and $P$) from the samples we *do* observe.

Estimating $P$ is especially challenging when $\Omega$ is continuous. In these cases, we are estimating $\Omega$ from a set with zero measure in $\Omega$. When $|\Omega|$ is finite, a sample of $\Omega$ may contain all possible outcomes, and thus the only unknown may be $P$. However, no finite sample can come close to exhausting all outcomes in $\Omega$ when $|\Omega|$ is infinite.

To make the estimation problem more tractable, we often assume that $P$ has a parametric form. Saying $P$ is parameterized by $\theta$ means we can know everything there is to know about $P$ from $\theta$. To estimate $P$, we first estimate (or *a priori* assume) $\Omega$, and then estimate $\theta$.

### D.2    Maximum Likelihood Parameter Estimation

The maximum likelihood parameter estimation procedure chooses parameter values that maximize the *likelihood function*: the conditional probability of the observed data on the parameters, $P(\mathbf{X}; \theta)$ (Murphy, 2012). The problem with maximum likelihood is that it is too "greedy." The maximum likelihood parameter estimation method does take into account the randomness associated with the assumed parametric form of the distribution, but not the randomness associated with choosing $\theta$ from $n$ samples. Consequently, maximum likelihood estimates are only guaranteed to be asymptotically unbiased and consistent (Johnson, 2013) but not unbiased for any $n$. Our method, PLE, seeks to incorporate the randomness associated with sampling $n$ times from a given parametrically defined probability function. This approach effectively de-biases the maximum likelihood estimate proportional to the uncertainty involved in the sampling process itself.

## E    One-Sided Uniform

### E.1    Maximum Likelihood Estimation of the One-Sided Uniform

Consider a $n$ samples drawn from the following uniform distribution

$$\mathbf{X} = [\mathbf{x}_1, \ldots, \mathbf{x}_n], \ \mathbf{x}_i \sim U[0, a] \ i = 1, \ldots, n.$$

We wish to estimate the parameter $\hat{a}$ from $\mathbf{X}$ so $\hat{a} = a$. First, we write out the likelihood function: $P_{\mathbf{X}|\hat{a}}(\mathbf{X}|\hat{a})$ as

$$P_{\mathbf{X}|\hat{a}}(\mathbf{X}|\hat{a}) = \begin{cases} 0 & \text{if } \hat{a} < \max(\mathbf{X}) \\ \frac{1}{\alpha \hat{a}^n} & \text{else} \end{cases}$$

Where $\alpha$ is a scaling factor that ensures the conditional PDF integrates to 1. Since this function is monotonic with respect to $\hat{a}$, the MLE is easily found as $\hat{a}_{\mathrm{MLE}} = \arg\max_{\hat{a}} P_{\mathbf{X}|\hat{a}}(\mathbf{X}|\hat{a}) = \max(\mathbf{X})$. This also corresponds to the $n$-th order statistic.

## E.2   Bias of the MLE of the One-Sided Uniform

Now that we have $\hat{a}_{\mathrm{MLE}}$, we can calculate the bias as follows:

$$b(\hat{a}_{\mathrm{MLE}}) = \mathbb{E}_{\mathbf{X}|a}[\hat{a}_{\mathrm{MLE}}] - a = \mathbb{E}_{\mathbf{X}|a}[\max(\mathbf{X})] - a \tag{7}$$

The expected value of the maximum of $\mathbf{X}$ (the $n$-th order statistic of $\mathbf{X}$) can be calculated by taking the derivative of the CDF of the maximum value with respect to the parameter in question:

$$F(\max(\mathbf{X})) = P(\max(\mathbf{X}) \leq \hat{a}) = \begin{cases} 0 & a < 0 \\ \left(\frac{\hat{a}}{a}\right)^n & \hat{a} \in [0, a] \\ 1 & \hat{a} > a \end{cases}$$

$$f(\max(X)) = P(\max(\mathbf{X}) = \hat{a}) \begin{cases} 0 & a < 0 \\ \frac{n\hat{a}^{n-1}}{a^n} & \hat{a} \in [0, a] \\ 0 & \hat{a} > a \end{cases}$$

Now we can calculate $\mathbb{E}[\max(\mathbf{X})]$ as follows:

$$\mathbb{E}_{\mathbf{X}|a}[\max(\mathbf{X})] = \frac{n}{a^n} \int_0^a \hat{a}^n d\hat{a} = \frac{n}{n+1} a. \tag{8}$$

Therefore, the bias of the MLE is

$$b(\hat{a}_{\mathrm{MLE}}) = \frac{n}{n+1} a - a = -\frac{1}{n+1} a.$$

Note that the bias here is negative, implying that the MLE $\hat{a}_{\mathrm{MLE}}$ will consistently *underestimate a*.

## E.3   One-Sided Uniform Example - Linear Function Class

Suppose we have data $\mathbf{X} = [\mathbf{x}_1, \ldots \mathbf{x}_n], \mathbf{x}_i \sim U[0, a], i = 1, \ldots, n$ and we want to estimate $\hat{a}$ from $\mathbf{X}$. We assume that $H$ is linear, so

$$\hat{a} = H(\mathbf{X}) = \sum_{i=1}^n \alpha_i \mathbf{x}_i$$

Now we calculate $H(\mathbf{Y})$, via

$$\mathbb{E}_{\mathbf{X},\mathbf{Y}}[H(\mathbf{Y}) - H(\mathbf{X})] = 0 \iff \mathbb{E}_{\mathbf{X},\mathbf{Y}}[H(\mathbf{Y})] = \mathbb{E}_{\mathbf{X}}[H(\mathbf{X})]$$

$$\text{First, we look at } \mathbb{E}_{\mathbf{X}}[H(\mathbf{X})]:$$

$$\mathbb{E}_{\mathbf{X}}[H(\mathbf{X})] = \sum_{i=1}^n \alpha_i \mathbb{E}[\mathbf{x}_i] = \mu \sum_{i=1}^n \alpha_i$$

$$\text{Now we look at } \mathbb{E}_{\mathbf{X},\mathbf{Y}}[sH(\mathbf{Y})]:$$

$$\mathbb{E}_{\mathbf{X},\mathbf{Y}}[sH(\mathbf{Y})] = \sum_{i=1}^n \alpha_i \mathbb{E}_{\mathbf{X},\mathbf{Y}}[\mathbf{y}_i] = \sum_{i=1}^n \alpha_i \cdot \left(\frac{\mu}{2} \sum_{j=1}^n \alpha_j\right) = \frac{\mu}{2} \left(\sum_{j=1}^n \alpha_j\right)^2$$

Combining these results gives:

$$\mathbb{E}_{\mathbf{X},\mathbf{Y}}\left[H(\mathbf{Y})\right] = \mathbb{E}_{\mathbf{X}}\left[H(\mathbf{X})\right] \implies \mu\left(\sum_{i=1}^{n}\alpha_i\right) = \frac{\mu}{2}\left(\sum_{j=1}^{n}\alpha_j\right)^2.$$

This equality only holds if $\sum_{j=1}^{n}\alpha_j = \frac{1}{2}\left(\sum_{j=1}^{n}\alpha_j\right)^2$. So $\sum_{j=1}^{n}\alpha_j$ must be 2 or 0. It can only sum to 0 if the mean is zero, otherwise $\mathbb{E}\left[H(\mathbf{X})\right] = 0$ always (this is the degenerate case). Thus, it must sum to 2. Also, $\alpha_i$ must be equal to $\frac{2}{n}$ because otherwise, permutations of the data would yield different results, and we are assuming they are independent. As a result,

$$\hat{a}_{\mathrm{PLE}} = H(\mathbf{X}) = \frac{2}{n}\sum_{i=1}^{n}\mathbf{x}_i = 2m_{\mathbf{X}}.$$

Where $m_{\mathbf{X}}$ is the sample mean, $m_{\mathbf{X}} = \frac{1}{n}\sum_{i=1}^{n}\mathbf{x}_i$. We calculate the bias by computing

$$\mathbb{E}_{\mathbf{X}|a}\left[\hat{a}_{\mathrm{PLE}}\right] = \mathbb{E}_{\mathbf{X}|a}\left[2\mathbb{E}\left[\mathbf{X}\right]\right] = 2\cdot\frac{a}{2} = a \implies b(\hat{a}) = a - a = 0.$$

Thus the PLE estimate of the one-sided uniform is unbiased.

### E.4  One-Sided Uniform Example - Function of the n-th order statistic

As before, we have a dataset $\mathbf{X} = [\mathbf{x}_1,\ldots\mathbf{x}_n], \mathbf{x}_i \sim U[0,a], i = 1,\ldots,n$, However this time, knowing the MLE estimate of $a$ is $\hat{a}_{\mathrm{MLE}} = \max(\mathbf{X})$, we assume the form of PLE is a linear function of the maximum of the data; i.e. $\hat{a}_{\mathrm{PLE}} = H(\mathbf{X}) = c\cdot\max\mathbf{X}$. We want to estimate $\hat{a}$ from $U[0,a]$. We calculate $H(\mathbf{Y})$, via $H(\mathbf{X})$, since $\mathbf{Y} \sim U[0, H(\mathbf{X})]$

We calculate $H(Y)$ as

$$\mathbb{E}_{\mathbf{X},\mathbf{Y}}\left[H(\mathbf{Y}) - H(\mathbf{X})\right] = 0 \implies \mathbb{E}_{\mathbf{X},\mathbf{Y}}\left[H(\mathbf{Y})\right] = \mathbb{E}_{\mathbf{X}}\left[H(\mathbf{X})\right]$$

Looking at $\mathbb{E}_{\mathbf{X},\mathbf{Y}}\left[H(\mathbf{Y})\right]$, we have

$$\mathbb{E}_{\mathbf{X},\mathbf{Y}}\left[H(\mathbf{Y})\right] = c\cdot\mathbb{E}_{\mathbf{Y}}\left[\max\mathbf{Y}\right] = c\cdot\frac{n}{n+1}\mathbb{E}_{\mathbf{X}}\left[H(\mathbf{X})\right]$$

Combining this with $\mathbb{E}_{\mathbf{X},\mathbf{Y}}\left[H(\mathbf{Y})\right]$, we have

$$\frac{cn}{n+1}\mathbb{E}_{\mathbf{X}}\left[H(\mathbf{X})\right] = \mathbb{E}_{\mathbf{X}}\left[H(\mathbf{X})\right] \implies c = \frac{n+1}{n}$$

As a result, $\hat{a}_{\mathrm{PLE}} = H(\mathbf{X}) = \frac{n+1}{n}\max\mathbf{X}$. We show the PLE estimate is unbiased by taking the expected value of this estimator,

$$\mathbb{E}_{\mathbf{X}|a}\left[\hat{a}_{\mathrm{PLE}}\right] = \mathbb{E}_{\mathbf{X}|a}\left[\frac{n+1}{n}\max\mathbf{X}\right] = \frac{n+1}{n}\cdot\frac{n}{n+1}a \implies \mathbb{E}_{\mathbf{X}|a}\left[\hat{a}_{\mathrm{PLE}}\right] = a,$$

Since $b(\hat{a}) = a - a = 0$), the PLE estimate of $a$ is unbiased. Furthermore, notice how we get the same result (in expectation) as the parametrization considered in section E.3. In other words, as long as different $H$'s contain the optimal value, PLE is transformation invariant and will select this optimal value regardless of the parametrization.

## F  Univariate Gaussian Parameters

### F.1  Maximum Likelihood Estimation

The probability density function for a univariate Gaussian random variable can be written as:

$$f(x) = \frac{1}{\sqrt{2\pi\sigma^2}}e^{-\frac{(x-\mu)}{2\sigma}}. \tag{9}$$

The likelihood function of a dataset $\mathbf{X}$ of $n$ such datapoints $x_i, i = 1, \ldots, n$ can be written as:

$$f(x_1, \ldots, x_n | \mu, \sigma) = \prod_{i=1}^{n} \frac{1}{\sqrt{2\pi\sigma^2}} e^{-\frac{(x_i - \mu)^2}{2\sigma^2}} \tag{10}$$

We find our maximum likelihood estimate of $\mu$ and $\sigma$ as

$$\mu_{\text{MLE}} = \arg \max_{\mu} f(x_1, \ldots, x_n | \mu)$$

and

$$\sigma_{\text{MLE}} = \arg \max_{\sigma} f(x_1, \ldots, x_n | \sigma).$$

Because we will be maximizing the likelihood, we can apply a logarithm (which is monotonic) to the likelihood function to get the log likelihood (ll) function:

$$ll(x_1, \ldots, x_n | \mu, \sigma) = \ln f(x_1, \ldots, x_n) = -\frac{n}{2} \ln 2\pi - \frac{n}{2} \ln \sigma^2 - \frac{1}{2\sigma^2} \sum_{i=1}^{n} (x_i - \mu)^2$$

First, the maximum likelihood of the mean can be found by taking the gradient of the above with respect to $\mu$ and solving when the gradient is 0:

$$\frac{\partial ll}{\partial \mu} = 0 \implies \frac{1}{\sigma^2} \sum_{i=1}^{n} (x_i - \mu) = 0 \implies \mu_{\text{MLE}} = \frac{1}{n} \sum_{i=1}^{n} x_i.$$

Similarly, we can calculate the maximum likelihood of the variance as

$$\frac{\partial ll}{\partial \sigma^2} = 0 \implies -\frac{n}{2\sigma^2} + \frac{1}{2(\sigma^2)^2} \sum_{i=1}^{n} (x_i - \mu)^2 = 0 \implies \sigma_{\text{MLE}}^2 = \frac{1}{n} \sum_{i=1}^{n} (x_i - \mu)^2.$$

Substituting in $\mu_{\text{MLE}}$ for $\mu$ in our MLE for $\sigma_{\text{MLE}}^2$, we have

$$\sigma_{\text{MLE}}^2 = \frac{1}{n} \sum_{i=1}^{n} \left( x_i - \frac{1}{n} \sum_{j=1}^{n} x_j \right)^2$$

$$= \frac{1}{n} \sum_{i=1}^{n} x_i^2 - \frac{2}{n^2} \sum_{i=1}^{n} \sum_{j=1}^{n} x_i x_j + \frac{1}{n^2} \sum_{i=1}^{n} \sum_{j=1}^{n} x_i x_j$$

$$= \frac{1}{n} \sum_{i=1}^{n} x_i^2 - \left( \frac{1}{n} \sum_{j=1}^{n} x_j \right)^2.$$

### F.2    Estimating the mean of a Gaussian with unknown parameters

Suppose $\mathbf{X} = [\mathbf{x}_1, \ldots, \mathbf{x}_n], \mathbf{x}_i \sim N(\mu, \sigma^2), i = 1, \ldots,$ and we wish to estimate $\mu$. Assume that $H$ is linear, so

$$\hat{\mu} = H(\mathbf{X}) = \sum_{i=1}^{n} \alpha_i \mathbf{x}_i$$

Now we solve for $H(\mathbf{Y})$

$$\mathbb{E}_{\mathbf{X}, \mathbf{Y}} [H(\mathbf{Y}) - H(\mathbf{X})] = 0 \iff \mathbb{E}_{\mathbf{X}, \mathbf{Y}} [H(\mathbf{Y})] = \mathbb{E}_{\mathbf{X}} [H(\mathbf{X})]$$

Looking at $\mathbb{E}_{\mathbf{X}, \mathbf{Y}} [H(\mathbf{X})]$, we have:

$$\mathbb{E}_{\mathbf{x}} [H(\mathbf{X})] = \sum_{i=1}^{n} \alpha_i \mathbb{E} [\mathbf{x}_i] = n\mu_{\mathbf{X}} \sum_{i=1}^{n} \alpha_i$$

Now we look at $\mathbb{E}_{\mathbf{X},\mathbf{Y}}[H(\mathbf{Y})]$:

$$\mathbb{E}_{\mathbf{x},\mathbf{y}}[H(\mathbf{y}_1,\ldots,\mathbf{y}_n)] = \sum_{i=1}^{n} \alpha_i \mathbb{E}[\mathbf{y}_i] = \sum_{i=1}^{n} \alpha_i \left(\mu_{\mathbf{X}} \sum_{j=1}^{n} \alpha_j\right)$$

Setting these two equal to each other, we have

$$\sum_{i=1}^{n} \alpha_i \left(\mu_{\mathbf{X}} \sum_{j=1}^{n} \alpha_j\right) = n\mu_{\mathbf{X}} \left(\sum_{j=1}^{n} \alpha_j\right)^2$$

The only way that this works is if $\sum_{j=1}^{n} \alpha_j = \left(\sum_{j=1}^{n} \alpha_j\right)^2$. So $\sum_{j=1}^{n} \alpha_j$ must be 1 or 0. It can only sum to 0 if the mean is zero, otherwise $\mathbb{E}[H(\mathbf{x}_1,\ldots,\mathbf{x}_n)] = 0$ always (this is the degenerate case). Thus, it must sum to 1. Also, $\alpha_i$ must be equal to $\frac{1}{n}$ because otherwise, permutations of the data would yield different results, and we are assuming they are independent. Thus,

$$\mu_{\text{PLE}} = \mu_{\mathbf{X}}.$$

Obviously, the sample mean is an unbiased estimator of the expected value, so this result is unbiased.

Since the above derivation does not use the Gaussian assumption, it can be repeated for any distribution to estimate its mean. Therefore, distributions whose mean characterize them are estimated unbiasedly with PLE. Such distributions include exponential, Bernoulli, Borel, Irwin–Hall, etc.

## F.3 Estimating the standard deviation of a Gaussian with unknown parameters

Now that we have the mean estimated, let us estimate the standard deviation. Since we used $H$ above to estimate the mean (which we now notate $H_\mu$), we will use $H_\sigma$ to estimate the variance to distinguish between the two. Suppose that $H_\sigma$ has the following quadratic form:

$$\hat{\sigma}^2 = H_\sigma(\mathbf{X}) = \sum_{i=1}^{n} \beta_i \left(\mathbf{x}_i - H_\mu(\mathbf{X})\right)^2 = \sum_{i=1}^{n} \beta_i \left(\mathbf{x}_i - \frac{1}{n}\sum_{j=1}^{n} \mathbf{x}_j\right)^2.$$

Now, we would like to evaluate $\mathbb{E}_{\mathbf{X}}[H_\sigma(\mathbf{X})]$ and $\mathbb{E}_{\mathbf{X},\mathbf{Y}}[H_\sigma(\mathbf{Y})]$ so that we can set them equal to each other and solve. Let us start with the former:

$$\mathbb{E}_{\mathbf{X}}[H_\sigma(\mathbf{X})] = \mathbb{E}_{\mathbf{X}}\left[\sum_{i=1}^{n}\beta_i\left(\mathbf{x}_i - \frac{1}{n}\sum_{j=1}^{n}\mathbf{x}_j\right)^2\right]$$

$$= \mathbb{E}_{\mathbf{X}}\left[\sum_{i=1}^{n}\beta_i\left(\mathbf{x}_i^2 - \frac{2}{n}x_i\sum_{j=1}^{n}\mathbf{x}_j + \frac{1}{n^2}\sum_{j=1}^{n}\sum_{k=1}^{n}\mathbf{x}_j\mathbf{x}_k\right)\right]$$

$$= \sum_{i=1}^{n}\beta_i\left(\mathbb{E}_{\mathbf{X}}[\mathbf{x}_i^2] - \frac{2}{n}\mathbb{E}_{\mathbf{X}}\left[\mathbf{x}_i\sum_{j=1}^{n}\mathbf{x}_j\right] + \frac{1}{n^2}\sum_{j=1}^{n}\sum_{k=1}^{n}\mathbb{E}_{\mathbf{X}}[\mathbf{x}_j\mathbf{x}_k]\right)$$

$$= \sum_{i=1}^{n}\beta_i\left(\left(\sigma^2+\mu^2\right) - \frac{2}{n}\left([\sigma^2+\mu^2]+[n-1]\mu^2\right) + \frac{1}{n^2}\left(n[\sigma^2+\mu^2]+[n^2-n]\mu^2\right)\right)$$

$$= \sum_{i=1}^{n}\beta_i\left(\left(1-\frac{2}{n}+\frac{1}{n}\right)\left(\sigma^2+\mu^2\right) - \left(\frac{2n-2}{n}-\frac{n-1}{n}\right)\mu^2\right)$$

$$= \sum_{i=1}^{n}\beta_i\left(\frac{n-1}{n}\left(\sigma^2+\mu^2\right) - \frac{n-1}{n}\mu^2\right)$$

$$= \frac{n-1}{n}\sum_{i=1}^{n}\beta_i\sigma^2.$$

On the other hand, we have

$$\mathbb{E}_{\mathbf{X},\mathbf{Y}}[H_\sigma(\mathbf{X})] = \frac{n-1}{n}\sum_{i=1}^{n}\beta_i\mathbb{E}[H_\sigma(\mathbf{X})] \qquad (*)$$

$$= \frac{n-1}{n}\sum_{i=1}^{n}\beta_i\left(\frac{n-1}{n}\sum_{j=1}^{n}\beta_j\sigma^2\right),$$

where $(*)$ is obtained by repeating the $\mathbb{E}_{\mathbf{X}}[H_\sigma(\mathbf{X})]$ derivation for $\mathbb{E}_{\mathbf{X},\mathbf{Y}}[H_\sigma(\mathbf{Y})]$. Setting these two equal to one another, we see that a necessary condition for $H[\mathbf{X}]_\sigma$ is that

$$\frac{n-1}{n}\sum_{i=1}^{n}\beta_i = 1.$$

Using similar arguments that we used with $H_\mu$, we see that $\beta_i = \beta_j$ for all $i,j \in \{1,\dots,n\}$. Therefore, we have that $\beta_i = \frac{1}{n-1}$ for all $i$, leading us to the unbiased MLE solution for the variance:

$$H(\mathbf{X}) = \frac{1}{n-1}\sum_{i=1}^{n}\left(\mathbf{x}_i - \frac{1}{n}\sum_{j=1}^{n}\mathbf{x}_j\right)^2.$$

If, instead, the mean $\mu$ is known and we do not have to solve for $H_\mu$, then $\beta_j = \frac{1}{n}$ and the proof is similar to the one in F.2.

## G   PLE is Asymptotically Unbiased

Suppose $\mathbf{X} \sim P_\theta$, and we wish to estimate $\theta$. Let $\mathbf{Y}$ be drawn from $P_{H(\mathbf{X})}$ so that $\mathbf{Y}$ is a new random variable which we want to satisfy the following equation

$$\mathbb{E}_{\mathbf{X},\mathbf{Y}}[H(\mathbf{Y}) - H(\mathbf{X})] = 0.$$

PLE is not as susceptible to MAD collapse when estimating Gaussian standard deviation

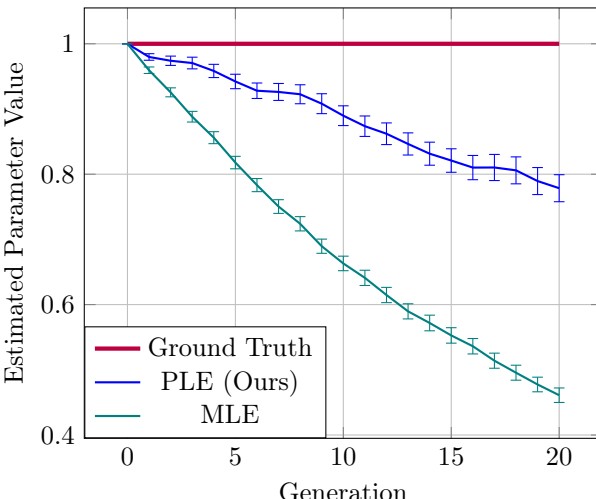

Figure 5: Generation index versus MLE and PLE of standard deviation for standard normal, $U[0,1]$. The error bars display the standard error for 100 different initializations. These results use the analytic form of the Gaussian standard deviation derived in Section F.3. For a data-driven version of this plot, see Figure 4.

This is equivalent to

$$\int_{\mathbf{y}} \int_{\mathbf{x}} H(\mathbf{y}) P(\mathbf{y}|H(\mathbf{x})) P(\mathbf{x}) d\mathbf{x} d\mathbf{y} = \int_{\mathbf{x}} H(\mathbf{x}) P(\mathbf{x}) d\mathbf{x} \tag{11}$$

Our ultimate goal is to calculate the bias. Let $s_x = H(\mathbf{X}) = \frac{1}{n} \sum_{i=1}^{n} H(\mathbf{x}_i)$ and $s_y = H(\mathbf{Y}) = \frac{1}{n} \sum_{i=1}^{n} H(\mathbf{y}_i)$.

$$\int_{\mathbf{y}} \int_{s_x} H(\mathbf{Y}) P\left(\mathbf{Y}|s_x\right) P\left(s_x|\theta\right) ds_x d\mathbf{Y} = \int_{s_x} s_x P(s_x|\theta) ds_x$$

$$\int_{s_y} \int_{s_x} s_y P\left(s_y|s_x\right) P\left(s_x|\theta\right) ds_x ds_y = \int_{s_x} s_x P(s_x|\theta) ds_x \tag{12}$$

We can thus say (in expectation)?

$$\mathbb{E}_{s_x;\theta} \left[ \int_{s_y} s_y P(s_y|s_x;\theta) ds_y \right] = \mathbb{E}_{\mathbf{X};\theta} \left[ s_x \right] \tag{13}$$

Thus when $s_x = \theta$,

$$\theta = \int_{s_y} s_y P(s_y|\theta) = \mathbb{E}[\hat{\theta}|\theta] \implies b(\hat{\theta}) = 0.$$

## H  Experimental Details

### H.1  Choosing $H$

As we show in E.1, as long as $\hat{\theta}_{\text{PLE}}$ (calculated from Equation 3) is in the domain of $H$, the choice of $H$ itself does not matter. In other words, PLE is transformation invariant for any transformation that could produce $\hat{\theta}_{\text{PLE}}$. In the absence of any prior information about

the form of $H$, the form of the MLE can be used as a reasonable prior for selecting $H$ and thus $\hat{\theta}_{\text{PLE}}$. In cases when MLE is strictly unbiased (not just asymptotically unbiased, as it is guaranteed to be (Johnson, 2013)), PLE will give the same result as MLE. At the very least, the MLE should be in the range of $H$, so when the MLE is unbiased, it is chosen.

## H.2 IMPLEMENTATION OF HYPERNETWORKS

In our implementation, we are able to automatically create a hypernetwork architecture given a generative model architecture. As long as the given architecture is a `torch.nn.Module`, our hypernetwork implementation outputs a named dictionary containing the layer names and weights for the target generative model. In addition, to allow for seamless usage of our PLE formulation as a drop-in replacement for existing objective functions for generative models, we exploit the abstractions provided by PyTorch that allow functional calls to any PyTorch `torch.nn.Module` that uses the predicted weights from our hypernetwork architecture. This allows us train existing generative models with PLE in just a few additional lines of code. More details regarding our implementation and code to reproduce the results can be found on GitHub[15].

## H.3 MADNESS EXPERIMENTS ON VARIOUS DISTRIBUTIONS

This section explains how the plots for Figure 4 were generated. Error bars show the standard error after either 100 or 1000 different initializations (some of the figures needed 1000 initializations for the error bars to decrease). Subfigure 1 shows the result from using the closed-form expression of PLE described in Section $E.4$, Subfigures 2-6 use the data-driven form from Equation 5, with 100 synthetic samples ($m = 100$) used to estimate the expectation in Equation 4.

Subfigure 1 (top-left) was generated from a one-sided Uniform distribution $\mathbf{X} \sim U[0, a]$ with true parameter $a = 1$, using $n = 20$ datapoints. The MLE is $a_{\text{MLE}} = \max \mathbf{X}$, which is derived in Section$E.1$, while $a_{\text{PLE}} = \frac{n+1}{n} \max(\mathbf{X})$, which is derived in Section E.4. The parameter $\hat{a}$ is estimated each iteration. Error bars show the standard error after 100 different initializations.

Subfigure 2 (top-middle) shows samples generated from a standard Gaussian (normal) distribution $\mathbf{X} \sim N[\mu, \sigma]$, with true parameters $\mu = 0, \sigma = 1$. The mean $\hat{\mu}$ and the standard deviation $\hat{\sigma}$ are estimated each iteration. We use $\mu_{\text{MLE}} = \frac{1}{n} \sum_{i=1}^{n} x_i$ and $\sigma_{\text{MLE}} = \frac{1}{n} \sum_{i=1}^{n} (x_i - \mu)^2$, which is derived in Section 5. The PLE estimates use the data-driven form from Equation 5, with 100 synthetic samples ($m = 100$). The estimates are generated with $n = 20$ points, and the results are averaged from 1000 initializations.

Subfigure 3 (top-right) shows samples generated from a Laplacian distribution $\mathbf{X} \sim$ Laplace$[\mu, b]$ with true parameters $\mu = 0, b = 1$. The mean $\mu$ and the scale parameter $b$ are estimated each iteration. The MLE of the parameters are $\mu_{\text{MLE}} = \text{median}(\mathbf{X})$ and $b_{\text{MLE}} = \frac{1}{n} \sum_{i=1}^{n} |x_i - \mu|$, and PLE estimates use the data-driven form from Equation 5, with 100 synthetic samples ($m = 100$). The estimates are generated with $n = 25$ points, and the results are averaged from 1000 initializations.

Subfigure 4 (bottom-left) shows samples generated from a Geometric distribution $\mathbf{X} \sim$ Geometric$[p]$, where the true parameter $p = 0.5$. The parameter $\hat{p}$ is estimated each iteration. The MLE of $p$ is $p_{\text{MLE}} = \frac{n}{\sum_{i=1}^{n} x_i}$, and PLE estimates use the data-driven form from Equation 5, with 100 synthetic samples ($m = 100$). The estimates are generated with $n = 25$ points, and the results are averaged over 1000 initializations.

Subfigure 5 (bottom-middle) shows samples generated from an Exponential distribution $\mathbf{X} \sim$ Exponential$[\lambda]$, where the true parameter $\lambda = 0.5$). The parameter $\hat{\lambda}$ is estimated each iteration. The MLE of $\lambda$ is $\lambda_{\text{MLE}} = \frac{n}{\sum_{i=1}^{n} x_i}$, and PLE estimates use the data-driven form

---

[15]Link is removed for double-blind review. Please refer to the code uploaded as supplementary material.

from Equation 5, with 100 synthetic samples ($m = 100$). The estimates are generated with $n = 25$ points, and the results are averaged over 1000 initializations.

Subfigure 6 (bottom-right) shows samples generated from a Type-I Pareto distribution $\mathbf{X} \sim \text{Pareto}[b]$, where the true parameter $b = 1.0$. The PDF of this distribution is $f(x, b) = \frac{b}{x^{b+1}}$, and $\hat{b}$ is estimated each iteration. The MLE of $b$ is $b_{\text{MLE}} = \frac{n}{\sum_{i=1}^{n}(\log(x_i)) - n\log(\min(\mathbf{X}))}$, and PLE estimates use the data-driven form from Equation 5, with 100 synthetic samples ($m = 100$). The estimates are generated with $n = 25$ points, and the results are averaged over 100 initializations.

The upper-left and upper-middle sub-figures show PLE estimated parameters slope down slightly. This is due to the fact that for a few runs, the variance goes to zero and cannot "recover" via a multiplicative constant. These degenerate runs bring the overall average down slightly, as there is no analogous degeneracy for large values. In essence, for the few estimates of the variance that are near zero, the result becomes clipped. This is sometimes described as variance collapse or model collapse in the literature (Alemohammad et al., 2023).

## I  EXTENDING FAIRNESS

While our definition of fairness considers only two classes of data (a majority and a minority class), this idea can easily be extended multi-class data. Suppose we have a dataset $\mathbf{X}$ with $n$ classes and a labeling of the data $\text{class}(\mathbf{x}) = y$. Consider a partitioning of $\mathbf{X}$ into each of these classes, where $\mathbf{X}_i = \{\mathbf{x} \in \mathbf{X} | \text{class}(\mathbf{x}) = i\}$   $i = 1, \ldots, n$ , where $\mathbf{X} = \mathbf{X}_1 \cup \mathbf{X}_2 \cup \cdots \cup \mathbf{X}_n$ and $|\mathbf{X}| = |\mathbf{X}_1| + |\mathbf{X}_2| + \cdots + |\mathbf{X}_n|$

This partitioning allows us to consider the imbalance ratios of data belonging to each class: Let $R_{i:j} = |\mathbf{X}_i|/|\mathbf{X}_j|$. We also consider the ratio score ratio between generated data from each class, $SR_{i:j} = S(M)_i/S(M)_j$ (note that with two classes, this is simply the fairness ratio between the majority and minority class indices). With multiple classes, the ratio of representation scores for a metric $M$ on generated data from two classes can be compared with the ratio of frequencies of data belonging to each class. This is a multi-class extension of the imbalance and fairness ratios described in Section 1.2.

Additionally, what is important to report is the generated freuqency of data belonging to each class. When the task is unconditional, an external classifier or oracle is needed to determine which class each generated datapoint belongs to. The frequency of generated data belonging to a given class should be compared to that class's frequency in the training data; i.e. if $\hat{\mathbf{X}}$ refers to synthetic data from a generative model and $\hat{\mathbf{X}}_i$   $i = 1, \ldots, n$ is a partitioning of the data according to its classified value, one should compare

$$|\hat{\mathbf{X}}_i|/|\hat{\mathbf{X}}| \lesseqgtr |\mathbf{X}_i|/|\mathbf{X}|$$

to determine if the generative model generates biased data according to the given classification.

## J  ABLATION EXPERIMENTS FOR PLE PENALTY

Our choice of $\lambda = 0.1$ was based on ablation experiments for the GMM example shown in Section 5.2. These experiments show that $\lambda = 0.1$ is the best choice; it performs better than MLE in the low-data regime and has less of a negative impact for larger samples than $\lambda = 0.0$ and $\lambda \geq 10$. Very large values of $\lambda$ cause the training to ignore the maximum likelihood term altogether which leads to poor performance.

Note that hyperparameter training provides an advantage over MLE even when the PLE penalty $\lambda$ is zero, as shown in Figure 6). We believe some of this benefit comes from averaging the weights from different batches, which is part of hyperparameter training shown in Equation 6. Work by Izmailov et al. (2019) has shown that averaging in weight space (as is done with evaluating the hypernetwork on batches of data) leads to better generalization and wider optima. As shown in the following figures, this averaging accounts for some (but not all) of the benefit of using our hypernetwork approach for training generative models.

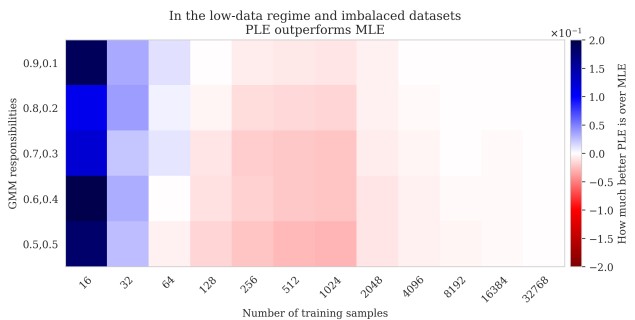

Figure 6: PLE vs MLE with $\lambda = 0.0$

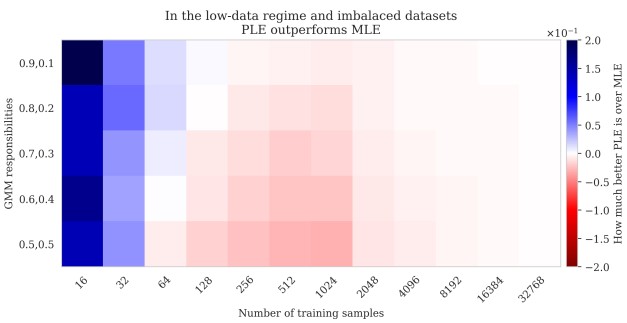

Figure 7: PLE vs MLE with $\lambda = 0.1$

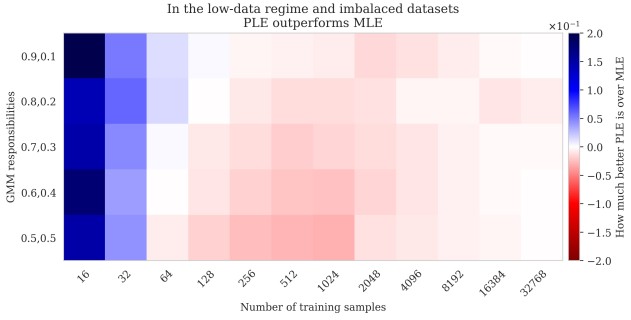

Figure 8: PLE vs MLE with $\lambda = 1.0$

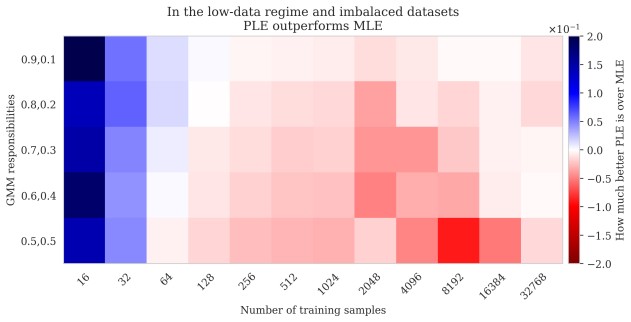

Figure 9: PLE vs MLE with $\lambda = 10.0$

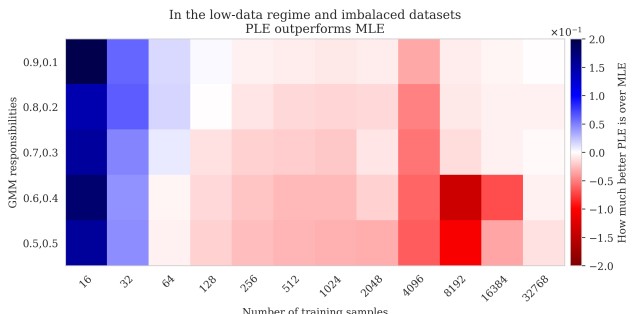

Figure 10: PLE vs MLE with $\lambda = 100.0$

Furthermore, our choice of hypernetwork architecture described in Section H.1 implicitly applies the PLE penalty during training. Below is a plot of training epoch versus estimated empirical bias, $\frac{1}{m} \sum_{i=1}^{m} \left| H(\hat{\mathbf{Y}}_i) - H(\mathbf{X}) \right|$ for both a naive hypernetwork architecture which features no averaging (Figure 11) and our proposed hypernetwork architecture (Figure 12). For both of these experiments, our hyperparameter $\lambda$ was chosen to be 0, so the empirical bias is not explicitly minimized. However our chosen hypernetwork structure implicitly minimizes the empirical bias during training to a certain extent. Increasing the value for $\lambda$ penalizes the empirical bias more, leading to models that are even more fair to datapoints belonging to minority classes and further stabilizing self-consumed estimation.

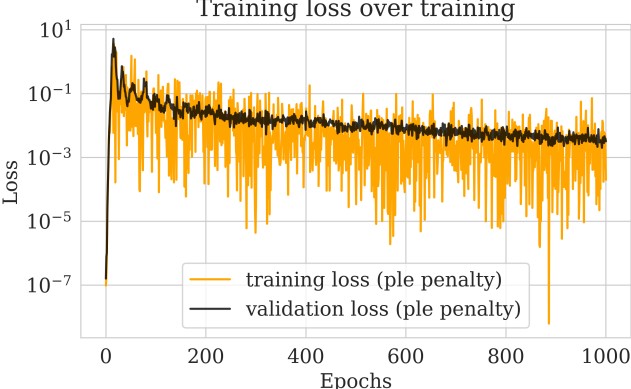

Figure 11: Training epoch vs. $\frac{1}{m} \sum_{i=1}^{m} \left| H(\hat{\mathbf{Y}}_i) - H(\mathbf{X}) \right|$, naive hypernetwork architecture and $\lambda = 0$

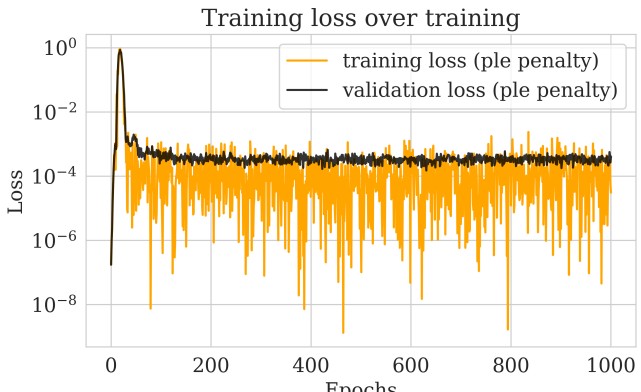

Figure 12: Training epoch vs. $\frac{1}{m}\sum_{i=1}^{m}\left|H(\hat{\mathbf{Y}}_i) - H(\mathbf{X})\right|$, proposed architecture and $\lambda = 0$

## K   GENERATED IMAGE EXAMPLES

Below are example images generated from the fairness experiments described in Section 5.1. Notice both the quality and quantity of minority images (Digit 6) are increased when training with the hypernetwork, shown in Figure, 13) versus standard training, shown in Figure 13).

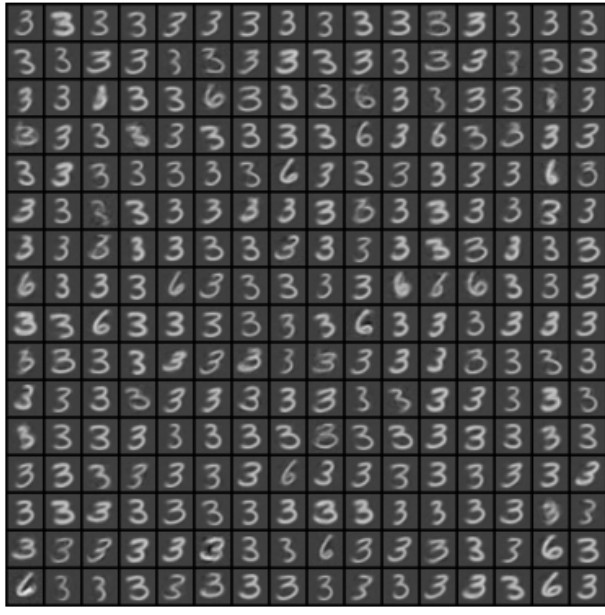

Figure 13: Sampled images from Hypernetwork VAE trained on subset of MNIST images with $R_I = 10 : 1$

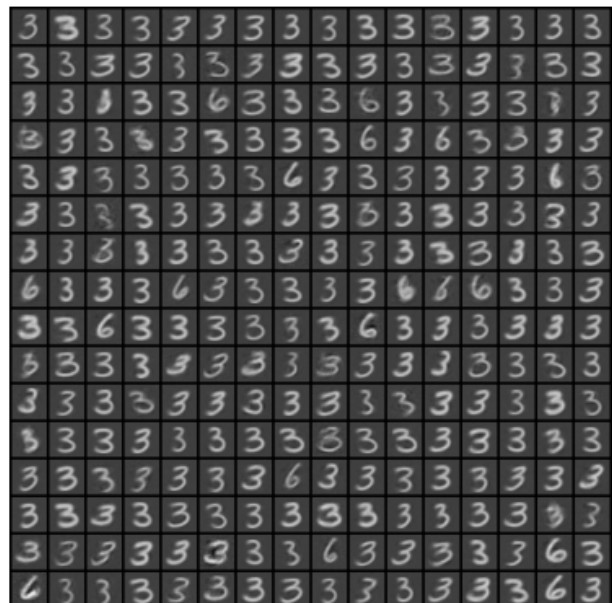

Figure 14: Sampled images from Vanilla VAE trained on subset of MNIST images with $R_I = 10 : 1$

