# OpenReview forum: "Improving Fairness and Mitigating MADness in Generative Models"
_ICLR.cc/2025/Conference — Submitted to ICLR 2025_

### Official Review · Reviewer_sR4f · 2024-11-01

**Soundness:** 2
**Presentation:** 3
**Contribution:** 2
**Rating:** 3
**Confidence:** 3

**Summary:**

The paper contains two new innovations:
- A metric of fairness when there is a minority and majority class in the training data.
- A hyperparameter network and corresponding loss function intended to prevent autophagous collapse (MADness)

The authors demonstrate the efficacy of their method to mitigate MADness over multiple generate-train loops and improve fairness.  They do this in both a deep generative model and a statistical context.

**Strengths:**

The proposed method is certainly interesting, and if it truly works as claimed it would be an impressive and efficient way to train new networks.  The paper is clear and well-motivated.  It poses an original solution to mitigate MADness in generative models.  The results regarding fairness seem experimentally compelling.

**Weaknesses:**

- Concerns about the validity of the method: Using a 3-layer FC neural network with the objective stated in the paper will likely result in pathologies.  Suppose that $\theta*$ is the MLE.  It is likely that the FC network H is learning to set its weights close to 0 and its biases close to $\theta*$.  In this way, changing the data is unlikely to alter the parameters very much, making any additional training/retraining using $H$ redundant.  One experiment that you could do to test this (which would alleviate my concerns and requires no additional training) is as follows.  Let G be the BigGAN from figure 2, and let H be the corresponding hyper-network.  If H is really doing it's job, then when you fit $\theta = \sum_{i=1}^n H(x_i)/n for x_i$ only in the category of "airplanes", the resulting generative network $G(x|\theta)$ should produce only images of airplanes.  I suspect that instead one of two things will happen: either the network will barely change, or it will produce images that are not discernible as anything.

- Quality concerns: although the authors report an FID score, there are no generated images from CIFAR in the paper or the appendix.  There also is no clear explanation for why there is a sudden spike in MADness at iteration 4.

- The evaluations are rather limited: The deep generative models are restricted to CIFAR-10.

- There is no theoretical explanation for why this method improves what the authors call fairness.  It is non-obvious to me why the method should improve fairness.

**Questions:**

See the weaknesses section.

Also:
- The definition of fairness seems overly simple-- it would be helpful to understand how statistics deals with this issue and what the existing metrics are.  For instance, how do you adapt the current definition when there are multiple classes?
- How does the method compare to more standard statistical techniques (i.e. upweighting the loss from the minority classes)?

---

> ### Author Response · Authors · 2024-11-23
>
> In response to weakness 1, we plan on running this test and will report back.
>
> In response to weakness 2, we added a new section to the appendix of the rebuttal revision (Section K) which includes generated images for some of the FID values provided.
>
> In response to weakness 3, our experiments were designed to show that our method works on 1) different generative models (VAE, GAN, GMM), 2) different datasets (MNIST, CIFAR, etc), and 3) architectures of varying complexity, from simple parameter estimation to BigGAN
>
> In response to weakness 2, the spike in MADness at iteration 4 occurs once the generations diverge enough from the training data.  We do not expect a linear increase in FID, and indeed madcow iteration vs the distance from the estimated to the true parameters is nonlinear for many distributions, as shown in Figure 4.   Also note that FID still increases, even with PLE; this collapse is observed for all generative models, see https://arxiv.org/abs/2307.01850 “Our primary conclusion across all scenarios is that without enough fresh real data in each generation of an autophagous loop, future generative models are doomed to have their quality (precision) or diversity (recall) progressively decrease”  We can hope to stabilize it and slow this down but this is observed for all generative models.
>
> In response to weakness 3, please note that we do train our method with a variety of other datasets beyond CIFAR-10, which includes multiple networks trained on MNIST in Section 5.1, GMM data in Section 5.2 (trained with a deep network).
>
> In response to claimed weakness 4, we describe theoretically why unbiased estimation improves fairness in lines 76-83, which has been reproduced here for the reviewer’s convenience:
> “When MLE produces biased estimates of the parameters (as it often does), the parameterized distribution becomes even more concentrated around existing high-probability events . Since probability distributions must integrate to 1, increasing the frequency of some events events comes at the expense of decreasing the frequency of others. The other events in this case are those that are less frequent, or belong to minority classes. As one can see in the first row in Figure H.3, biased maximum likelihood estimates eventually collapse towards the mode(s) of the data and thus will underrepresent data away from the mode. As a result, unbiased estimators will generate distributions that more accurately represent the frequency of minority events.”
>
> In response to question 1, we added a new section in the rebuttal revision (Section I) that explains how our definition of fairness can easily be extended to data belonging to multiple classes.  It introduces a few new metrics for evaluating the between-class scores and frequencies, including a new metric that reports the frequency of generated data (separate from the quality of such data)

---

> > ### Author Response · Authors · 2024-11-23
> >
> > In response to question 2, we added a new section in the rebuttal revision that describes how our work differs from existing fairness methods in Section 1.2, where we also mention a few other statistical methods.  The relevant section reads:
> > > While recent work has looked at improving fairness in generative models, our work differs conceptually in its focus on removing statistical bias. By removing statistical bias, we avoid over-representing data belonging to majority classes without needing to specify any protected attributes or classes. Other approaches are either restricted to a single model type or require data labeled with protected attributes. For instance, FairGAN (Xu et al., 2018) proposes a variant of GANs that requires labeled data with protected attributes, and can only be used for training GANs. Choi et al. (2020) proposes a method that uses two datasets in situations when a smaller dataset may better represent the population ratios, but does not address bias in the learning process itself.
> >
> > >The gradient clipping approach suggested by Kenfack et al. (2022) seeks to improve fairness by biasing the dataset towards uniformity. They write that their goal is “to improve the ability of GAN models to uniformly generate samples from different groups, even when these groups are not equally represented in the training data.” This differs from our model in that 1) it actively biases the model to favor more uniform generation of points with different classes to achieve fairness, and 2) requires data labeled with protected attributes, which may not be feasible to expect. Finally, Rajabi and Garibay (2022) suggest a method for generating tabular data whose statistics match a reference dataset. This approach also relies on explicit labels of the protected attribute in the training dataset and is restricted to GANs. Our method can be used for any generative model and requires no labels or information about the protected attribute
> >
> > Upweighting the loss, which is similar to the gradient clipping method, will bias the generations in favor of uniformly generating data according to a given protected attribute.  Our proposed method is superior in two ways: 1) it does not require introducing statistical bias; in fact it removes it, and 2) our method does not require data labeled according to a protected attribute, which would be required for upweighting the loss.

---

> > > ### Comment · Reviewer_sR4f · 2024-11-26
> > > **Rebuttal Response 1**
> > >
> > > My apologies for the slow response.
> > >
> > > >In response to weakness 2, we added a new section to the appendix of the rebuttal revision (Section K) which includes generated images for some of the FID values provided.
> > >
> > > Why are these images restricted to MNIST?  Don't you think it would be better if you showed CIFAR-10 too and illustrated the progression of MADness for your method versus the original method?  Most likely you already have these images given that you performed FID evaluations.
> > >
> > > > In response to weakness 3, our experiments were designed to show that our method works on 1) different generative models (VAE, GAN, GMM), 2) different datasets (MNIST, CIFAR, etc), and 3) architectures of varying complexity, from simple parameter estimation to BigGAN
> > >
> > > I understand that resources can be scarce (especially if you are an academic lab), but my point was that CIFAR-10 is essentially a toy dataset by modern standards.  Your response to this objection was to say that you fit the model on two other toy datasets.  I was hoping to see something of greater scale either in terms of data cardinality or image resolution-- my apologies if this was unclear in my review.
> > >
> > > >“When MLE produces biased estimates of the parameters (as it often does), the parameterized distribution becomes even more concentrated around existing high-probability events . Since probability distributions must integrate to 1, increasing the frequency of some events events comes at the expense of decreasing the frequency of others. The other events in this case are those that are less frequent, or belong to minority classes. As one can see in the first row in Figure H.3, biased maximum likelihood estimates eventually collapse towards the mode(s) of the data and thus will underrepresent data away from the mode. As a result, unbiased estimators will generate distributions that more accurately represent the frequency of minority events.”
> > >
> > > I remain unconvinced.  You are using a hyper-network to fit the parameters of the network, but the loss function to fit the hyperparameter network is still a constrained MLE.  It's really unclear to me why this should reduce the bias.  Is there any mathematical reason?
> > >
> > > I like the idea of this paper, but based on the results in this paper and the lack of mathematical proof, I'm still unconvinced that it actually works.  For the time being, I will keep my current score.  This could change depending on the outcome of experiment intended to address Weakness 1.

---

### Official Review · Reviewer_f4tT · 2024-11-04

**Soundness:** 2
**Presentation:** 2
**Contribution:** 2
**Rating:** 5
**Confidence:** 2

**Summary:**

This paper addresses fairness and bias in generative models, which often penalize minority classes and suffer from model autophagy disorder (MADness). The authors propose training generative models with hypernetworks to make them more fair, stable, and less biased. They introduce a regularization term to reduce discrepancies between a model's performance on real data versus synthetic data, helping mitigate bias and improve representation fairness. The experimental results show that their framework is scalable, supporting integration with existing deep generative models such VAEs and GANs.

**Strengths:**

+ The overall paper is well-structured and clearly written, with concise explanations that make concepts like hypernetworks and MADness accessible to a broader readers.

+ This paper tackles an important and timely issue by addressing fairness and bias in generative models, particularly focusing on challenges such as minority class penalization and model autophagy disorder (MADness).

+ The discussion related to large language models (LLMs) adds practical relevance to this research, as the content generated by LLMs is widely distributed across the internet.

**Weaknesses:**

- The experiments are relatively weak, especially given the small dataset and the older models used (VAE and BigGAN). It would strengthen the paper significantly if the method were tested on diffusion models.
- There is extensive discussion of ChatGPT and other LLMs in Section 1.3. It would enhance the paper if the proposed method could be applied directly to LLMs.

**Questions:**

1. In Section 1.1, why does it say that GAN is trained with MLE?

2. In Line 087-088, should $ R_I = C_{Maj} / C_{Min} $ actually be $ R_I = |C_{Maj}| / |C_{Min}| $?

3. In Equation 2, why is $S(M)$ linear? Could the authors provide a detailed explanation for this?

4. In Line 316, how should we interpret $ R_{Fair} < |C_{Maj}| / |C_{Min}| $? According to your conclusion about the linearity of $ S(M) $, shouldn’t the optimal $ R_{Fair} $ that fits the data be equal to $ |C_{Maj}| / |C_{Min}| $?

5. How are the hypernetworks $H_\phi$ trained?

---

> ### Author Response · Authors · 2024-11-22
>
> In response to weakness 1, we agree that training on diffusion models is important, however the expense of sampling from diffusion models means future work is needed to adapt our method to diffusion models.  We explain this in the conclusion, which reads, “Since hypernetwork training involves sampling the generative model to evaluate the penalty, future work is needed to allow tractable training of diffusion models, which are expensive to sample from.”  Additionally, in a new section of the appendix, we explain how this method could still provide a benefit to diffusion models without this overhead, which reads “There is likely still a benefit to using hypernetworks for training diffusion models due to the averaging operation that occurs, which is known to convexify the loss function, see See Averaging Weights Leads to Wider Optima and Better Generalization, https://arxiv.org/abs/1803.05407)”
>
> In response to weakness 2, our experiments were designed to show that our method works on 1) different generative models (VAE, GAN, GMM), 2) different datasets (MNIST, CIFAR, etc), and 3) architectures of varying complexity, from simple parameter estimation to BigGAN.  While our work can be applied to LLMs directly, it is out of the scope of our current paper for space reasons.
>
> In response to question 1, we say GANs are trained with MLE because, as Goodfellow describes, “[t]the training objective for D can be interpreted as maximizing the log-likelihood for estimating the conditional probability” (See https://arxiv.org/abs/1406.2661).  We cite Vapnik’s connection to empirical risk minimization and log likelihood maximization with maximum likelihood estimation on line 53.
>
> In response to question 2, yes this was a typo and has been fixed in the rebuttal revision, thank you for pointing out.
>
> In response to question 3, the assumption of linearity for S(M) was to illustrate that the loss incurred for data belonging to the majority class contributes more to the overall loss than that incurred for data belonging to the minority class.  If the reviewer thinks this illustration is potentially confusing in its assumption of linearity, we can remove this equation.
>
> Your question about line 316 and whether R_Fair should be less than or equal to |CMaj|/|Cmin| is one we thought about as well.  RFair<|CMaj|/|Cmin| refers to a case where the model’s performance does not scale linearly with class frequencies, as the score disparity between majority and minority data would actually be *less* than expected from the ratio of frequencies.  On one hand, one can certainly argue that RFair<|CMaj|/|Cmin| is unfair to majority datapoints, the same way RFair<|CMaj|/|Cmin| is unfair to minority datapoints.  However, in the context of optimization, data belonging to the same class is likely close in feature space.  As a result, decreasing the loss from data from one class will improve the performance of data in that class on average more than those not belonging to this class.  This can cause saturation, and the terms that contribute to the loss most eventually become data that is not as well represented.  This smaller distance between points of the same class explains why RFair can be pushed below the imbalance ratio without the model itself being biased.  If the number (not the quality) of minority datapoints to majority points generated (classified by an external oracle) differed from the ratio, this would be problematic and indicative of bias.
>
> In response to Question 5, we explain how our hypernetworks are trained in Section H.2 and in Section 4 (The formatting of the equations was getting messed up when trying to copy it in the comments) We also plan on providing our code upon publication (the Github link has been removed during the blind section of the review process).

---

### Official Review · Reviewer_je7K · 2024-11-04

**Soundness:** 2
**Presentation:** 3
**Contribution:** 2
**Rating:** 3
**Confidence:** 3

**Summary:**

The paper sets out to address the important problem of representation bias in generative models and MADness (decreased models’ performance when trained using their outputs). The authors propose a method that enforces the parameters of the generative model to remain consistent when trained on the original or synthetic data. To alleviate the challenges of intractable search over generative models’ parameters trained on different synthetic data, the proposed method uses hypernetworks to sample generative model weights. Experiments show the ability of the proposed method to mitigate unfairness and MADness.

**Strengths:**

1. The paper is well-written and addresses important problems in generative models.
2. The proposed method is intuitive and easy to understand.
3. The experiments are well conducted and demonstrate the effectiveness of the method.

**Weaknesses:**

The paper lacks discussion and comparisons with existing bias mitigation methods in generative models. The authors could consider the following methods and explain how their method differs conceptually from these existing approaches and provide empirical comparisons that can support the benefit of the proposed method in the considered setup:

- Xu, Depeng, et al. "Fairgan: Fairness-aware generative adversarial networks." 2018 IEEE international conference on big data (big data). IEEE, 2018.
- Choi, Kristy, et al. "Fair generative modeling via weak supervision." ICML 2020.
- Sabbagh, Kamil, et al. "RepFair-GAN: Mitigating Representation Bias in GANs Using Gradient Clipping." Tiny Papers @ ICLR (2023).
- Rajabi, Amirarsalan, and Ozlem Ozmen Garibay. "Tabfairgan: Fair tabular data generation with generative adversarial networks." Machine Learning and Knowledge Extraction 4.2 (2022): 488-501

The experiments are not consistent enough; specifically, the datasets used in the paper are considered for different types of experiments. For example, fairness is evaluated only on MNIST using VAE, and MADness mitigation is evaluated on CIFAR 10 using GANs. The authors should provide justifications for the experimental design or consider evaluating different datasets under the same set of experiments, i.e., show that the method also mitigates MADness/fairness on MNIST with GANs, and similarly for CIFAR 10. This would demonstrate the generalizability of the proposed method under the metric being evaluated (e.g., fairness or MADness).

Another important concern is the authors did not provide sufficient justification or insights into how the method can mitigate bias in generated data, particularly when a generative model trained on the original data is already biased, which can be exacerbated in the follow-up generation. More specifically, it is unclear how constraining the MLE to find the model parameters that remain consistent with synthetic and original data (Eq. 4) mitigates unfairness. The authors could provide a more detailed explanation or intuition for how their method addresses bias, particularly in cases where the original training data is biased.

The authors did not provide ablation on the impact of the parameter $\lambda$ in Equation 5. As the regularization term proposed is the core contribution of the paper, providing these experiments would provide more insights into how the regularization term influences several aspects of the work: the quality of the generated data, the number of generations after which “MADness” occurs, and the fairness of the synthetic data. These experiments could be provided for synthetic distributions used in Figure 4.  In addition, the authors can discuss the range of $\lambda$ values they think would be most interesting to explore and why.

The Appendix contains unnecessary materials (e.g., B, C, D, E). While this material is interesting and well-written, it can confuse the reader since it is not directly linked to the paper's main contribution. Instead, the authors should consider referring the readers to books/papers that contain this background information.

For the experiments on the MNIST dataset, Line 334 reads: _The majority class was the digit 3, and the minority class was the digit 6 (this choice was arbitrary)_. Instead of arbitrarily choosing the minority class, the authors should consider the class with the highest false negative rate when classified as the minority. This means the minority class confuses the most with other classes and can be harder to learn when it is underrepresented, for example, see how reference [1] chooses the class to artificially under-represent.

The paper highly depends on hypernetworks, which increase the complexity of the method and its practical usage.

[1]Bagdasaryan, Eugene, Omid Poursaeed, and Vitaly Shmatikov. "Differential privacy has disparate impact on model accuracy." NeurIPS 2019

**Questions:**

Please see the weaknesses above. In addition, the paper needs proofreading to fix minor typos. Here is a non-exhaustive list of them:
- Line 060: Autophogy Estimation => Autophagy
- Line 078: [...] some events events[...] => some events
- Line 088: [...] and is is described => and is described
- Line 322: [...] S ompares the representation => S compares the representation
- Line 414: For hyeprnetwork training [...] => For hypernetwork

---

> ### Author Response · Authors · 2024-11-22
>
> In response to the first weakness, we added a new section that describes how our work differs from existing fairness methods (the papers suggested refer to fairness and not bias as we are describing it).   This can be found in Section 1.2 of the Rebuttal Revision, and is copied here below:
> > While recent work has looked at improving fairness in generative models, our work differs conceptually in its focus on removing statistical bias. By removing statistical bias, we avoid over-representing data belonging to majority classes without needing to specify any protected attributes or classes. Other approaches are either restricted to a single model type or require data labeled with protected attributes. For instance, FairGAN (Xu et al., 2018) proposes a variant of GANs that requires labeled data with protected attributes, and can only be used for training GANs. Choi et al. (2020) proposes a method that uses two datasets in situations when a smaller dataset may better represent the population ratios, but does not address bias in the learning process itself.
>
> >The gradient clipping approach suggested by Kenfack et al. (2022) seeks to improve fairness by biasing the dataset towards uniformity. They write that their goal is “to improve the ability of GAN models to uniformly generate samples from different groups, even when these groups are not equally represented in the training data.” This differs from our model in that 1) it actively biases the model to favor more uniform generation of points with different classes to achieve fairness, and 2) requires data labeled with protected attributes, which may not be feasible to expect. Finally, Rajabi and Garibay (2022) suggest a method for generating tabular data whose statistics match a reference dataset. This approach also relies on explicit labels of the protected attribute in the training dataset and is restricted to GANs. Our method can be used for any generative model and requires no labels or information about the protected attribute
>
> In response to the second weakness, our experiments were designed to show that our method works on 1) different generative models (VAE, GAN, GMM), 2) different datasets (MNIST, CIFAR, etc), and 3) architectures of varying complexity, from simple parameter estimation to BigGAN.
>
> In response to the third weakness, training a generative model with hypernetworks allows bias to be removed via parametric bootstrapping, as described in section 3.2.  How parametric bootstrapping removes bias is found in Hall’s 1992 book The Bootstrap and Edgeworth Expansion.  The relationship between bias removal or correction and fairness is described in Section 1.2.  The relevant section reads:
> > .Recent work has shown that generative models carry and often amplify
> unbalances present in training data (Zhao et al., 2018). When MLE produces biased estimates
> of the parameters (as it often does), the parameterized distribution becomes even more
> concentrated around existing high-probability events3. Since probability distributions must
> integrate to 1, increasing the frequency of some events comes at the expense of decreasing the
> frequency of others. The other events in this case are those that are less frequent, or belong
> to minority classes. As one can see in the first row in Figure H.3, biased maximum likelihood
> estimates eventually collapse towards the mode(s) of the data and thus will underrepresent
> data away from the mode. As a result, biased estimators will learn distributions where
> majority-class data is overrpresented and minority-class data is underrepresented, while
> unbiased estimators will learn distributions that more accurately represent the frequency of
> minority events.
> While recent work has looked at improving fairness in generative models, our work differs
> conceptually in its focus on removing statistical bias. By removing statistical bias, we avoid
> over-representing data belonging to majority classes…
> The scope of our work does not consider cases where the original training data is biased, as this knowledge could only come from additional (training) data or an oracle.  Appropriate data collection (e.g. random sampling) is needed to ensure training data is unbiased.  Instead we look at whether the model learns a biased representation from the given data.

---

> ### Author Response · Authors · 2024-11-22
>
> For your comment about lambda, we added a section (Section J in the rebuttal revision) which expands on our choice of lambda = 0.1, which we chose empirically via ablation experiments.  The relevant section reads:
> > “Our choice of λ = 0.1 was based on ablation experiments for the GMM example shown in Section 5.2. These experiments show that λ = 0.1 is the best choice; it performs better than MLE in the low-data regime and has less of a negative impact for larger samples than λ = 0.0 and λ ≥ 10. Note that hyperparameter training provides an advantage over MLE even when the PLE penalty is zero, as shown in Figure 6). We believe this benefit comes from averaging the weights from different batches, which is part of hyperparameter training shown in Equation 6. Work by Izmailov et al. (2019) has shown that averaging in weight space (as is done with evaluating the hypernetwork on batches of data) leads to better generalization and wider optima. As shown in the following figures, this averaging accounts for some (but not all) of the benefit of using our hypernetwork approach for training generative models. Furthermore, our choice of hypernetwork architecture described in Section H.1 implicitly applies the PLE penalty during training. Below is a plot of training epoch versus estimated empirical bias, for both a naive hypernetwork architecture which features no averaging (Figure 11) and our proposed hypernetwork architecture (Figure 12).
>
> > For both of these experiments, our hyperparameter λ was chosen to be 0, so the empirical bias is not explicitly minimized. However our chosen hypernetwork structure implicitly minimizes the empirical bias during training to a certain extent. Increasing the value for λ penalizes the empirical bias more, leading to models that are even more fair to datapoints belonging to minority classes and further stabilizing self-consumed estimation.”
>
> In response to the fourth weakness about the relevance of the appendix, each section of the appendix is referenced in the main body of the paper.  We reference Appendix A, which discusses hybrid Bayesian and Freqnetist methods when discussing the connection of our work to parametric bootstrapping in Section 2.1, as Efron’s work also connects parametric bootstrapping to hybrid Bayesian and frequentist methods.  Section B is referenced in a footnote in Section 1.2 for readers unfamiliar with Kolmogorov’s axiomatization of probability theory, which is used for statistical estimation.  Our theory in Section 1.2 refers to events which rely on this axiomatization.  Appendix C provides an overview of generative models, and is referenced in Section 2.1 when discussing generative model training and unbiased estimation.  Appendix D describes Maximum Likelihood in more detail, which forms the basis for the motivation for our method.  Section E is cited both in Section 5.2 illustrating our method for closed-form solutions for distributions and in Section 4 to motivate our hypernetwork architecture.  Sections E and F are both used to find closed-form solutions to distributions shown in Figure 4 in Section 5.  Section G shows our method is asymptotically unbiased, illustrating our claim about bias.  Section H describes our implementation details for Figure 4 in Section 5 and provides more details on our hypernetwork implementation described in Section 4.  The additional appendix sections have been added in response to other reviewer comments and were not present in the initial submission.
>
> For your comment on line 334, we added an additional sentence explaining our choice of digits 3 and 6.  The goal of the experiment is to show the effect of our method on the minority class, which depends primarily on the frequency of occurrence and not the class itself.  Any two digits from the dataset could be chosen as long as the training frequencies differ by the reported amount.  We thus chose these digits at random.
>
> For the final weakness, you claim, “The paper highly depends on hypernetworks, which increase the complexity of the method and its practical usage.”  As we describe in lines 487-489, the computational overhead is minimal.  We believe this is in part due to the weight averaging which convexifies the loss function (See Averaging Weights Leads to Wider Optima and Better Generalization, https://arxiv.org/abs/1803.05407).  The benefits of increased fairness and robustness to MADness justify the additional overhead (with the overhead being minimal, as stated in our paper).  For example, the FairGAN paper you referenced above justifies training  additional discriminators based on the increased performance on minority data.
>
> We have fixed all typos mentioned, thank you.

---

> ### Comment · Reviewer_je7K · 2024-11-27
> **Response to Rebuttal**
>
> I thank the authors for their detailed response. However, the rebuttal mostly stops at repeating what was mentioned in the paper without addressing most of the concerns raised. I highlight a few of the issues below.
>
> > our experiments were designed to show that our method works on 1) different generative models (VAE, GAN, GMM), 2) different datasets (MNIST, CIFAR, etc), and 3) architectures of varying complexity, from simple parameter estimation to BigGAN.
>
> The issue was to demonstrate the robustness of the solution by performing the same set of experiments across different datasets and generative models. Using a single and different dataset to evaluate each aspect of the work does not make the conclusions sufficiently robust and might hide some limitations.
>
> > In response to the third weakness, training a generative model with hypernetworks allows bias to be removed via parametric bootstrapping, as described in section 3.2 [...] The scope of our work does not consider cases where the original training data is biased, as this knowledge could only come from additional (training) data or an oracle. Appropriate data collection (e.g. random sampling) is needed to ensure training data is unbiased. Instead we look at whether the model learns a biased representation from the given data.
>
> It remains unclear to me how the proposed method can mitigate bias and improve fairness. Furthermore, if the original training data is not biased, which is rarely the case, it is unclear what form of representation bias the paper aims to study.  Furthermore,  the paper studies the effect of the proposed method on the minority class and randomly underrepresents certain classes on the MNIST dataset, which makes the dataset biased. This makes the statement "_The scope of our work does not consider cases where the original training data is biased, as this knowledge could only come from additional (training) data or an oracle_" inconsistent with the methodology of the paper; please clarify if I missed some part of the work.
>
> > The benefits of increased fairness and robustness to MADness justify the additional overhead (with the overhead being minimal, as stated in our paper). For example, the FairGAN paper you referenced above justifies training additional discriminators based on the increased performance on minority data.
>
> The authors should provide compelling arguments/proofs on how the proposed method increases fairness and also justify why the method assumes the training data is unbiased. As of now, I maintain my score.

---

### Official Review · Reviewer_LyWA · 2024-11-04

**Soundness:** 3
**Presentation:** 1
**Contribution:** 3
**Rating:** 5
**Confidence:** 3

**Summary:**

The paper proposes penalized autophogy estimation (PLE), which is a method that impproves fairness and reduce MADness of generative models.
The key idea is to make the model recursively stable via a regularization term on the MLE loss.
A naive formulation of this loss is intractable and the author(s) solve this issue by a hypernetwork that generates the parameters of the generative model
Experiments show that the proposed method can improve fairness and reduce MADness of generative models on various datasets.

**Strengths:**

- Novel theoretical contribution connecting statistical bias in MLE to fairness and MADness issues in generative models
- Comprehensive empirical validation across multiple types of distributions and models (VAE, BigGAN)
- Strong technical foundation with clear connections to existing statistical theory
- Results show meaningful improvements in both fairness metrics and stability against MADness

**Weaknesses:**

- The motivation and problem setup in the introduction is not well structured, making it difficult to grasp the core contribution initially
- Limited ablation studies on the choice of hyperparameters (e.g., PLE penalty $\lambda$=0.1)
- Some experimental results show inconsistent trends across different distributions without sufficient explanation
- The presentation could be more accessible to readers less familiar with statistical estimation theory

**Questions:**

1. How sensitive is the method to the choice of $\lambda=0.1$? Was this value chosen empirically or is there theoretical justification?
2. In Figure 2, why does FID still increase with PLE, albeit more slowly? Is this related to the hyperparameter choice (e.g. number of data points in equation 6)?
3. In Figure 4, some distributions show stable performance while others show increasing MADness---why?
4. How does the hypernetwork architecture choice and hyperparameters like $\lambda$ affect performance? I think we need a systematic study on this?

In general I really like the paper and I'd like to increase my rating if the presentation is improved and my question on hyperparameter is answered.

---

> ### Author Response · Authors · 2024-11-22
>
> In response to weakness 1, the accessibility of our work is important because we want to ensure people can understand it, however the motivation for our method does come directly from estimation theory.  Sections B-F in the appendix provide a background on topics from estimation theory relevant to our contribution.
>
> In response to weakness 2 and questions 1 and 4, about our choice of $\lambda$, we added a section (Section J in the rebuttal revision) which expands on our choice of lambda = 0.1, which we chose empirically via ablation experiments.  It reads:
> >“Our choice of λ = 0.1 was based on ablation experiments for the GMM example shown in Section 5.2. These experiments show that λ = 0.1 is the best choice; it performs better than MLE in the low-data regime and has less of a negative impact for larger samples than λ = 0.0 and λ ≥ 10. Note that hyperparameter training provides an advantage over MLE even when the PLE penalty is zero, as shown in Figure 6). We believe this benefit comes from averaging the weights from different batches, which is part of hyperparameter training shown in Equation 6. Work by Izmailov et al. (2019) has shown that averaging in weight space (as is done with evaluating the hypernetwork on batches of data) leads to better generalization and wider optima. As shown in the following figures, this averaging accounts for some (but not all) of the benefit of using our hypernetwork approach for training generative models. Furthermore, our choice of hypernetwork architecture described in Section H.1 implicitly applies the PLE penalty during training. Below is a plot of training epoch versus estimated empirical bias, for both a naive hypernetwork architecture which features no averaging (Figure 11) and our proposed hypernetwork architecture (Figure 12).
>
> >For both of these experiments, our hyperparameter λ was chosen to be 0, so the empirical bias is not explicitly minimized. However our chosen hypernetwork structure implicitly minimizes the empirical bias during training to a certain extent. Increasing the value for λ penalizes the empirical bias more, leading to models that are even more fair to datapoints belonging to minority classes and further stabilizing self-consumed estimation.”
>
> In response to question 2, FID still increases, even with PLE; this collapse is observed for all generative models, see https://arxiv.org/abs/2307.01850 “Our primary conclusion across all scenarios is that without enough fresh real data in each generation of an autophagous loop, future generative models are doomed to have their quality (precision) or diversity (recall) progressively decrease”  We can hope to stabilize it and slow this down but this is observed for all generative models.
>
> In response to question 3, for Figure 4, the MLE for all six distributions collapses.  The reason PLE slightly diverges from the true value for a few of the plots (in particular the standard normal) is due to the variance dropping to zero for a few degenerate runs.  A new section has been added to the appendix describing Figure 4 in H.3 to clarify, which reads:
> The upper-left and upper-middle sub-figures show PLE estimated parameters slope down slightly.  This is due to the fact that for a few runs, the variance goes to zero and cannot “recover'” via the multiplicative constant added in the chosen form. These degenerate runs bring the overall average down slightly, as there is no analogous degeneracy for large values.  In essence, for the few estimates of the variance that are near zero, the result becomes clipped.  This is sometimes described as variance collapse or model collapse in the literature (See pages 8 and 17 of https://arxiv.org/abs/2307.01850)
>
> In response to question 4, section J in the rebuttal revision (as mentioned above) contains more details on how $\lambda$ is chosen and why our specific hypernetwork choice is beneficial, especially in comparison to naive hypernetworks.
> Regarding the comment on our choice of the hypernetwork architecture, as described in Section 4, our design choice is based on the following three requirements: (i) permutation-invariance with respect to input data, as the generative model weights should not depend on the ordering of input data; (ii) the ability to estimate the generative model weights given an arbitrary number of input data, which is essential for batch training; and (iii) reducing computational overhead, which we achieve in Equation (6), since the inner sum is embarrassingly parallel and the outer network is a collection of independent fully connected layers that get evaluated in parallel.  We agree that future work can study the impact of different hyperparameter architectures for overall performance.
>
> In response to weakness 3, we want to be as consistent as possible, so please provide specific details about what is inconsistent in our results so that we may better address them and have a clearer paper.  Thank you.

---

### Meta-Review · Area_Chair_9FwT · 2024-12-19

**Metareview:**

The paper aims to address the problem of representation bias in generative models, which is the models’ decreased performance when trained using their own synthetic outputs. The main idea is to use a regularization to make sure that model parameters are stable during training. The authors also proposed to use a hypernetwork to tackle the intractable regularization. The reviewers are concerned about the lack of both conceptual and empirical comparisons against existing methods in the literature of generative modeling that achieve the same goal. Furthermore, the evaluation could be significantly improved both qualitatively (by showing the models' generated images) as well as quantitatively (by comparing against other baseline methods). The authors responded to the reviewers' questions but they do not address the above concerns.

The authors are encouraged to take the reviewers' comments to strengthen the work further.

**Additional Comments On Reviewer Discussion:**

Reviews remain unchanged.

---

### Decision · Program_Chairs · 2025-01-22

Reject